# Switch of cell migration modes orchestrated by changes of three-dimensional lamellipodium structure and intracellular diffusion

Chao Jiang[1,2,3], Hong-Yu Luo[1,3], Xinpeng Xu[4,5], Shuo-Xing Dou [1,3], Wei Li[1,6], Dongshi Guan [3,7], Fangfu Ye[1,3,6,8], Xiaosong Chen [2], Ming Guo [9], Peng-Ye Wang [1,3,6] ✉ & Hui Li [2] ✉

Cell migration plays important roles in many biological processes, but how migrating cells orchestrate intracellular molecules and subcellular structures to regulate their speed and direction is still not clear. Here, by characterizing the intracellular diffusion and the three-dimensional lamellipodium structures of fish keratocyte cells, we observe a strong positive correlation between the intracellular diffusion and cell migration speed and, more importantly, discover a switching of cell migration modes with reversible intracellular diffusion variation and lamellipodium structure deformation. Distinct from the normal fast mode, cells migrating in the newly-found slow mode have a deformed lamellipodium with swollen-up front and thinned-down rear, reduced intracellular diffusion and compartmentalized macromolecule distribution in the lamellipodium. Furthermore, in turning cells, both lamellipodium structure and intracellular diffusion dynamics are also changed, with left-right symmetry breaking. We propose a mechanism involving the front-localized actin polymerization and increased molecular crowding in the lamellipodium to explain how cells spatiotemporally coordinate the intracellular diffusion dynamics and the lamellipodium structure in regulating their migrations.

Cell migration is a highly complex process in which diverse translocations and interactions of intracellular molecules orchestrate cellular morphologies and behaviors in time and space[1–3]. It plays important roles in morphogenesis, wound healing, and tumor metastasis[4,5]. Previous studies of cell migration have been conducted extensively on different aspects of intracellular signaling pathways and cytoskeletal structures[3,6–15], cellular morphologies and mechanics[16–22], and extracellular matrixes and environmental topology[23–28]. However, the role of intracellular diffusion in cell migration remains unclear.

Diffusion provides the physical basis for most molecular transports in cells and thus mediates many important biological functions, including diffusion-controlled reactions, signal transduction, and phase separation[29–31]. In dynamically migrating cells, intracellular diffusion is even more essential. For example, in lamellipodium

[1]Beijing National Laboratory for Condensed Matter Physics and Laboratory of Soft Matter Physics, Institute of Physics, Chinese Academy of Sciences, Beijing 100190, China. [2]School of Systems Science and Institute of Nonequilibrium Systems, Beijing Normal University, Beijing 100875, China. [3]School of Physical Sciences and School of Engineering Sciences, University of Chinese Academy of Sciences, Beijing 100049, China. [4]Physics Program, Guangdong Technion—Israel Institute of Technology, 241 Daxue Road, Shantou, Guangdong 515063, China. [5]Technion—Israel Institute of Technology, Haifa 32000, Israel. [6]Songshan Lake Materials Laboratory, Dongguan, Guangdong 523808, China. [7]State Key Laboratory of Nonlinear Mechanics, Institute of Mechanics, Chinese Academy of Sciences, Beijing 100190, China. [8]Wenzhou Institute, University of Chinese Academy of Sciences, Wenzhou, Zhejiang 325001, China. [9]Department of Mechanical Engineering, MIT, 77 Massachusetts Ave, Cambridge, MA 02139, USA. ✉e-mail: pywang@iphy.ac.cn; huili@bnu.edu.cn

protrusions during cell migration, actin filaments push the cell membrane forward by a treadmilling mechanism, in which the maintenance of continuous actin polymerization at the front lamellipodium requires the replenishment of actin subunits from the rear lamellipodium where actin depolymerization occurs[3,32–34]. The active fluid flow with a relatively slow speed cannot account for an intracellular transport that matches the cell speed[22,35]. A detailed description of the intracellular diffusion as well as its correlation with molecular distributions and dynamics underlying different cell migrative behaviors, is still missing. On the other hand, intracellular diffusion is also strongly influenced by molecular crowding and subcellular structures[36–44]. When cells alter their migration behaviors, the corresponding biophysical changes in the lamellipodium would have a profound effect on the intracellular diffusion dynamics. However, it remains unclear how a migrating cell coordinates intracellular diffusion with subcellular structures in space and time to drive and regulate its migration. Obviously, a multiscale investigation of cell migration in view of intracellular diffusion could provide us with new information to decipher the underlying mechanism of complex cellular behaviors from collective molecular dynamics.

Here, by simultaneously monitoring the intracellular diffusion and the 3D lamellipodium structure in single migrating fish keratocyte cells, we have observed a positive correlation between the intracellular diffusion rate and the cell migration speed, which is coupled with the deformation changes of the lamellipodium. Moreover, we have observed a left-right symmetry-breaking of both intracellular diffusion and lamellipodium structure in cells that are turning directions. A mechanism involving the front-localized actin polymerization and increased molecular crowding in the lamellipodium is proposed to explain the correlation between the intracellular diffusion, the 3D lamellipodium structure, and cell migration behavior.

## Results

### Cell migration speed is positively correlated with intracellular diffusion

We use fish keratocytes, one of the simplest and most classic model systems, to study cell migration[10,16,22,27]. These cells consist of a broad and flat lamellipodium in the front and a round cell body in the rear (Fig. 1a). Fish keratocytes mainly make two types of motions: mostly persistent migrations and occasionally short turns (Fig. S1).

To investigate the intracellular diffusion dynamics of fish keratocytes, we use fluorescent quantum dots (QDs) coated with polyethylene glycol (PEG) as the nonspecific diffusing probes, with a size of around 30 nm[22,39,45]. The QDs are loaded into keratocytes through the osmotic lysis of pinocytic vesicles[46], then, their diffusions in single migrating cells are monitored by single particle tracking (SPT)[39,47] (Fig. 1a, b). These QDs are proven to be located in the cytosol but not trapped in the vesicles (Fig. S2). In the reference of cell frame in which the cell movement is removed[22,32], the QD trajectories cover the whole lamellipodium and part of the cell body (Fig. 1c, d), with typical trajectories shown in Fig. 1e. An analysis of the mean square displacement (MSD) shows that the QDs diffuse fast in the lamellipodium region with a diffusion exponent of $\alpha \approx 0.98$ and a diffusion rate of $D \approx 0.82$ μm²/s, but much more slowly in the cell body with $\alpha \approx 0.74$ and $D \approx 0.36$ μm²/s (Fig. 1f). These results indicate that the cell body is highly crowded, probably due to the compacted cellular organelles therein[31,36,47]. In the lamellipodium, however, more free spaces are available for facilitating more rapid translocations of QDs whose sizes are comparable to biomacromolecules such as actin subunits[22]. Given that the lamellipodium is the main site where actin recycling and protrusions take place[34], such a less crowded intracellular environment is crucial for efficient macromolecule translocations during cell migration. Moreover, even isolated lamellipodia have also been shown to be able to migrate spontaneously and efficiently[34]. It is, therefore, quite interesting to elucidate the physical properties and intracellular dynamics of

lamellipodia, especially their spatial and temporal evolution in migrating cells.

Interestingly, we observe a strong positive correlation between the cell migration speed and the QD diffusion rate in single cells. An example is given in Fig. 1g–k, where the cell speed increases and then decreases while $D$ undergoes dynamic changes following the same trend. By analyzing over 66 keratocyte cells, the strong positive correlation is further confirmed (Fig. 1l), and moreover, there is a considerable (5-fold) variation in both $D$ (0.5–2.5 μm²/s) and the cell migration speed (0.05–0.25 μm/s). We have eliminated the possible influence of cell migration on intracellular diffusion (Fig. S3). Moreover, we find that the speed of intracellular flow is slower than the lower limit of our detection and thus has no detectable influence on the measured intracellular diffusion rates (Fig. S4)[22].

It has been previously reported that keratocyte shape is correlated with cell speed[16], which we indeed have confirmed. But more interestingly, we find new correlations between intracellular diffusion and cell migration behaviors: positive correlations between lamellipodium area and cell speed and between lamellipodium area and $D$ (Fig. S5). These results suggest potential functional roles of intracellular diffusion in cell motility.

### A new migration mode is revealed by excluding gap regions in diffusion maps in slow cells

To further probe the spatial characteristics of intracellular diffusion in cells with different migration speeds, we investigate the intracellular diffusion map of QDs through the whole intracellular space of single migrating cells. To make such a map for a cell, $D$ is calculated using the segments of QD trajectories near each position[39]. We find that in a fast cell, the diffusion of QDs is present across the whole lamellipodium (Fig. 2a), and in a slow cell, however, a gap area appears in the rear of its lamellipodium, indicating that the diffusing QDs are excluded from this area. Correspondingly, the QD trajectories also clearly show that almost all the QDs are compartmentalized within the front lamellipodium in a slow cell (Fig. 2b). In fact, the excluded gap region at the rear lamellipodium is still clearly observed in an aligned diffusion map from different cells, demonstrating that the compartmentalization is general in slow cells (Fig. S6). We further observe that all the cells exhibiting such an excluded gap area are prone to be slower than those without a gap (Fig. S7). Moreover, to eliminate the possible effect of biochemical property and size of the probes on the intracellular diffusion, we use 500- and 70-kD dextrans which are widely known as non-specific probes[39,45], with their sizes comparable to and smaller than the QDs, respectively. They both show a similar positive correlation between the intracellular diffusion coefficient and cell speed, as well as the compartmentalized diffusion in the front lamellipodia of slow cells (Fig. S8).

Intracellular diffusion is tightly coupled with subcellular structures[36,39,44]. We speculate that the presence of strikingly excluded gap regions in the lamellipodia of slow cells is resulting from a physical barrier formed when a cell reduces its speed. To further investigate this phenomenon, we use 5-chloromethylfluorescein diacetate (CMFDA), a small fluorescence probe staining the cytoplasm, to indicate cell morphology[48]. The thickness of lamellipodium is proportional to the fluorescence intensity of CMFDA under epifluorescence imaging[22]. As expected, we find that in fast cells, the fluorescence intensity is uniform throughout, indicating a flat lamellipodium, and in slow cells, however, a clearly discontinuous gap shows up in the rear of the lamellipodium near the cell body (Fig. 2c). Especially, the normalized CMFDA intensity in the front lamellipodium of slow cells is obviously higher than that in the flat lamellipodium of fast cells. These results suggest that in slow cells, the lamellipodium swells up in the front but significantly thins down in the rear. Moreover, we find no differences in the cell volume or in the nuclei volume and height between fast and slow cells (Fig. S9). Considering the coincidence of the QD-excluded

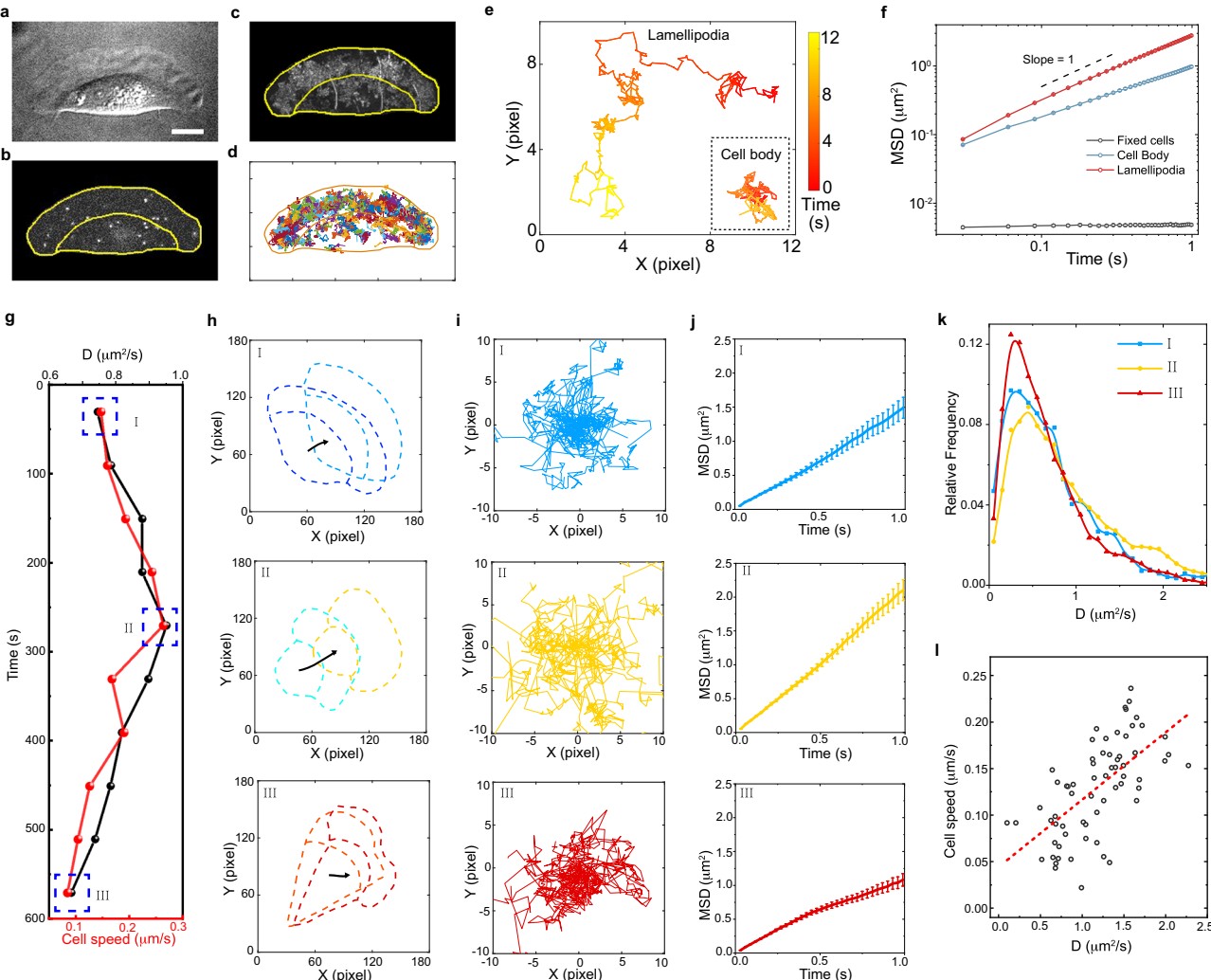

**Fig. 1 | Cell migration speed correlates with intracellular diffusion. a–d** Cell loaded with QDs: a migrating keratocyte (**a**), QDs loaded in the cell (**b**), overlay of QD movies in the reference of cell frame (**c**) and QD trajectories (**d**). The upper and lower regions in each figure correspond to the lamellipodium and the cell body, respectively. Scale bar, 10 μm. **e** Typical QD trajectories in the lamellipodium and the cell body, showing distinct dynamic diffusion behaviors. **f** MSD plots of QDs in two sub-regions of migrating cells ($n = 10$) and in fixed cells ($n = 7$). Slopes of MSD curves are 0.98 and 0.74 for lamellipodium (total trajectories, 1863) and cell body

region (total trajectories, 686), respectively. **g** Cell speed and QD diffusion rate $D$ as a function of time for a single migrating cell. **h–j**, Cell displacements (**h**), QD trajectories for 50 frames (**I**), and MSD plots (**j**) at three time points (I–III, indicated in (**g**)). Data are presented as mean ± SE. **k** Distributions of $D$ at the three-time points with averages at 0.72, 0.99, and 0.64 μm²/s, respectively. The trajectory numbers are 115, 275, and 148 for I–III, respectively. **l** The average migrating speed of each cell is plotted against the average $D$ in its lamellipodium for a total of 66 cells. Source data are provided as a Source Data file.

region (Fig. 2f) with the CMFDA-depleted region (Fig. 2g), it can then be understood why QDs are excluded from the rear lamellipodia in slow cells. In normal fast-migrating keratocytes, the lamellipodia are already very thin (~250 nm)[49,50]. Thus with the thickness of the gap region in the rear lamellipodium of slow cells further attenuated, QDs with large sizes (~30 nm) are readily squeezed out of the rear and accumulate in the front lamellipodium, whereas smaller CMFDA molecules may still occupy the whole lamellipodium. That is, the large QDs are separated from small CMFDA in the lamellipodia of slow cells. Consistent with this, when the migrating cell slows down, we have observed the rapid translocation of intracellular QDs from the rear part of lamellipodium to the front part, taking place within ~3 s (Fig. S10).

We next check whether the diffusion of small molecules is modified in the swollen lamellipodia of slow cells. Since the diffusion of small CMFDA molecules is too fast to be tracked, we perform the fluorescent recovery after photobleaching (FRAP). It is observed that the CMFDA intensity recovers more slowly in the swollen lamellipodia of slow cells, compared with that in the normal lamellipodia of fast

cells (Fig. 2h), further demonstrating the positive correlation between intracellular diffusion and cell speed.

The lamellipodia of keratocytes are mostly constituted of and shaped by actin filaments[3,18,32]. It is, therefore, reasonable to hypothesize that the lamellipodium structural changes are driven by the actin filaments. To justify this hypothesis, we apply stimulated emission depletion (STED) microscopy to explicitly image the organization of actin filaments labeled by SiR-actin (Fig. 2d). As expected, the actin filaments are uniformly and normally distributed in the flat lamellipodia of fast cells, whereas more actin filaments are observed in the front but fewer in the rear of uneven lamellipodia of slow cells (Fig. S11 and S12a), consistent with previous results measured for the altered actin networks in cells with reduced speed by mechanical loading[18]. In addition to the actin filaments, we image the distribution of unpolymerized actin monomers labeled by Alexa594-Deoxyribonuclease I. Interestingly, in contrast to the uniform distribution in lamellipodia of fast cells, more actin monomers are observed in the front lamellipodia of slow cells (Fig. S12b), which is consistent with the distribution of

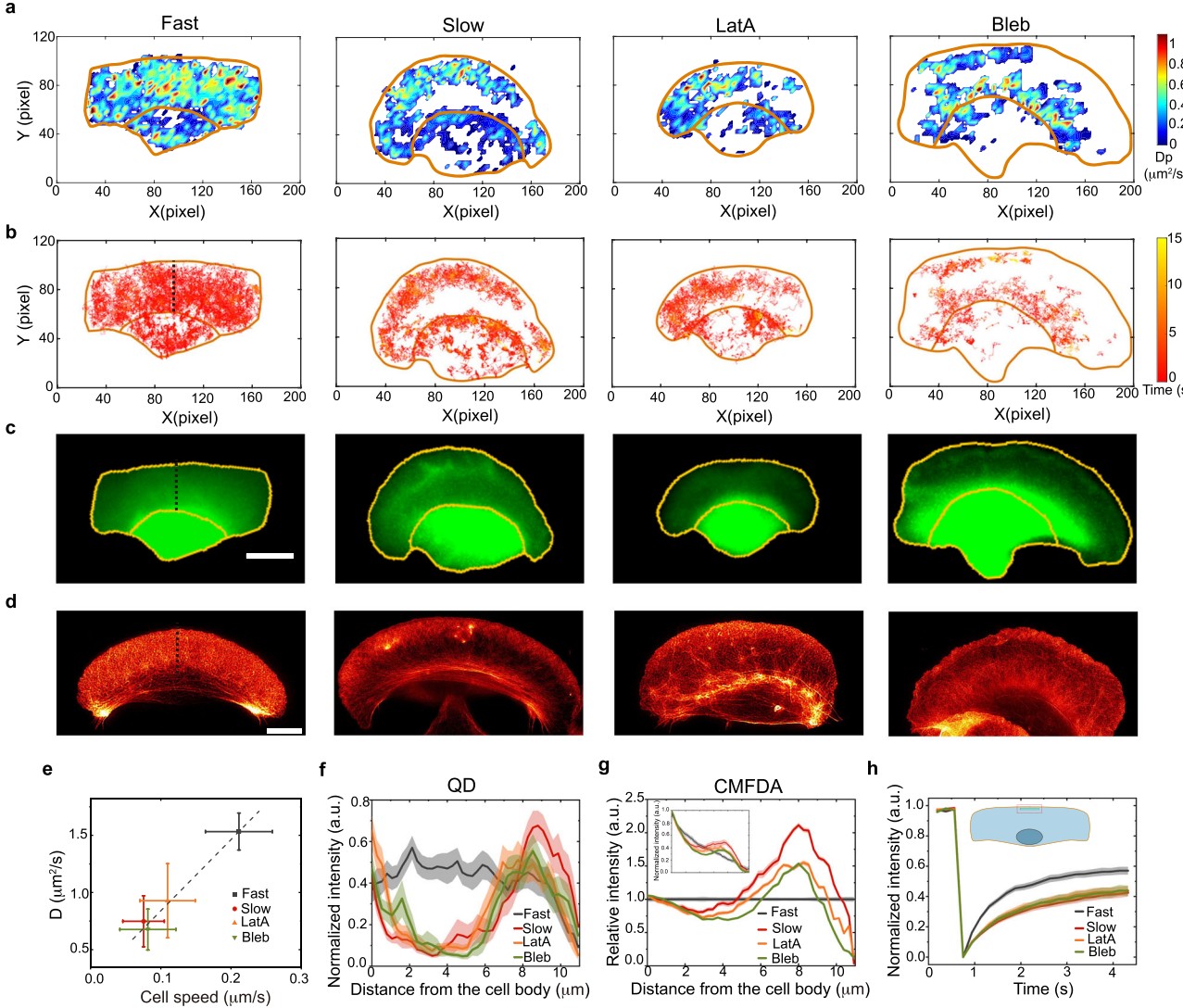

**Fig. 2 | Fast and slow migrating cells show distinct intracellular diffusion maps and lamellipodium structures. a–c** QD diffusion maps (**a**), QD trajectories (**b**), and CMFDA fluorescence images (**c**) for migrating cells with fast and slow speeds and for cells treated with 10 nM latrunculin A (LatA) or 50 μM blebbistatin (Bleb). The color range from blue to red corresponds to $D_p$ (local diffusion rate) from 0 to 0.8 μm²/s. The cells are to move upwards, with their outlines and cell body regions labeled by yellow lines. **d** STED imaging of actin filaments labeled by SiR-actin in fixed cells at the four indicated cell conditions. **e** Average $D$ and migration speeds for cells at four conditions (fast, $n = 16$; slow, $n = 16$; LatA, $n = 15$; Bleb, $n = 14$). Data are shown as mean ± SD. **f, g** Normalized intensity profiles of QD movie projection (**f**) and CMFDA (**g**) along a line across the lamellipodium, from the cell body to its leading edge (dotted line). **h** FRAP of the CMFDA intensity in the front lamellipodium at four conditions, with the fitted recovery times: 0.73 ± 0.18 s (fast, $n = 35$), 1.09 ± 0.37 s (slow, $n = 40$), 1.06 ± 0.30 s (LatA, $n = 19$) and 1.04 ± 0.25 s (Bleb, $n = 20$). The solid lines and the shaded regions represent mean ± SE. Inset schematic for the FRAP region in the front lamellipodium. Scale bar, 10 μm. Source data are provided as a Source Data file.

actin filaments and coincides with the lamellipodium volume indicated by CMFDA.

Collectively, the above results reveal a new keratocyte migration mode with a slow speed, which is characterized by a deformed lamellipodium with reduced intracellular diffusion and compartmentalized large molecules in the front. These results also imply that the molecular diffusion and structural changes inside the lamellipodia may play key roles in regulating the migration speed of the keratocytes.

## The reversible transition between fast and slow migration modes is coupled with changes in intracellular dynamics and lamellipodium structure

To further examine whether the lamellipodium structural changes are generally occurring in keratocytes when their migration speed is reduced, we interfere with cell migration speed by several pharmacological methods[16]. Firstly, we perturb the actin polymerization rate

by adding latrunculin A (LatA). Secondly, as actomyosin contraction generates mechanical forces in cell contraction and migration, we inhibit the myosin II activity by adding blebbistatin (Bleb). Notably, we observe similar changes in the diffusion of both QDs and CMFDA as observed for cells spontaneously switching from fast mode to slow mode (Fig. 2a, b). And both the actin filaments and monomers show redistribution after the drug treatments, consistent with that in the slow mode (Fig. 2d and Fig. S12). When the cells' migration speeds are reduced, $D$ decreases (Fig. 2e), and both the QD-excluded region of the diffusion map and the corresponding gap area of the CMFDA image appear in the rear lamellipodium (Fig. 2f, g). Moreover, a previous study has shown that the actin flow can be effectively suppressed by the simultaneous inhibition of action polymerization and myosin contractility[32]. We find that the treatments with Bleb and LatA, or with Bleb and the actin stabilization drug jasplakinolide (Jasp), show similar results as those for slow cells (Fig. S13).

The above pharmacological interferences with cell speed are functioning by reducing the membrane-advancing speed since both actin protrusions and actomyosin contractions provide essential mechanical forces that push the membrane forward[1,3]. In addition, we notice that when a keratocyte is not fully isolated from a monolayer, the leading cell is under mechanical loading by other cells via intercellular connections[18,27], which then also reduces the speed of the leading cell. In this case, we also observe an excluded region in the QD diffusion map and a corresponding gap area in the CMFDA image (Fig. S14). Furthermore, the redistribution of actin networks (Fig. S15) is found to be consistent with previous reports that the actin filament organization in migrating cells is changed to adapt to external mechanical loading[18]. When we tether a microneedle to the cell body and thus apply a backward drag force, the cell also switches from the fast migration mode to the slow migration mode with a gap region appearing in the lamellipodium (Fig. S16).

We then examine whether the switching of cell migration mode is reversible if the membrane-advancing speed is increased back. Interestingly, when we remove the above drugs (Fig. S17) or add Calyculin A to enhance myosin II activities (Fig. 3a–c), we indeed observe a reversible switching from the slow mode back to the normal fast mode.

Moreover, similar reverse switching is observed for cells that just escape from the cell monolayer and get rid of the external mechanical loading (Fig. 3d, e). In these experiments, the lamellipodium resumes its flat shape.

Next, we sought to tune the intracellular diffusion rate by changing molecular crowding in cells via the osmotic pressure[48,51]. The addition of PEG 300 in the medium would increase the external osmotic pressure. To match the external osmotic pressure, the cell volume decreases accordingly (Fig. S9). Interestingly, it is observed that the addition of 1% PEG 300 has made the cell switch from the fast to the slow modes, with the QD-excluded region in the lamellipodium area emerging accordingly (Fig. 3f–h). Conversely, when the medium is replaced by water, a reversible switch from slow mode to fast mode occurs (Fig. S18). This experiment further demonstrates the coordination between the intracellular diffusion and lamellipodium structure during the reversible switch of cell migration modes.

### Lamellipodium thickness profiles are measured at nanometer resolution by using 3D single-particle tracking

The above cell volume images have given an estimation of the lamellipodium thickness. To precisely determine the lamellipodium

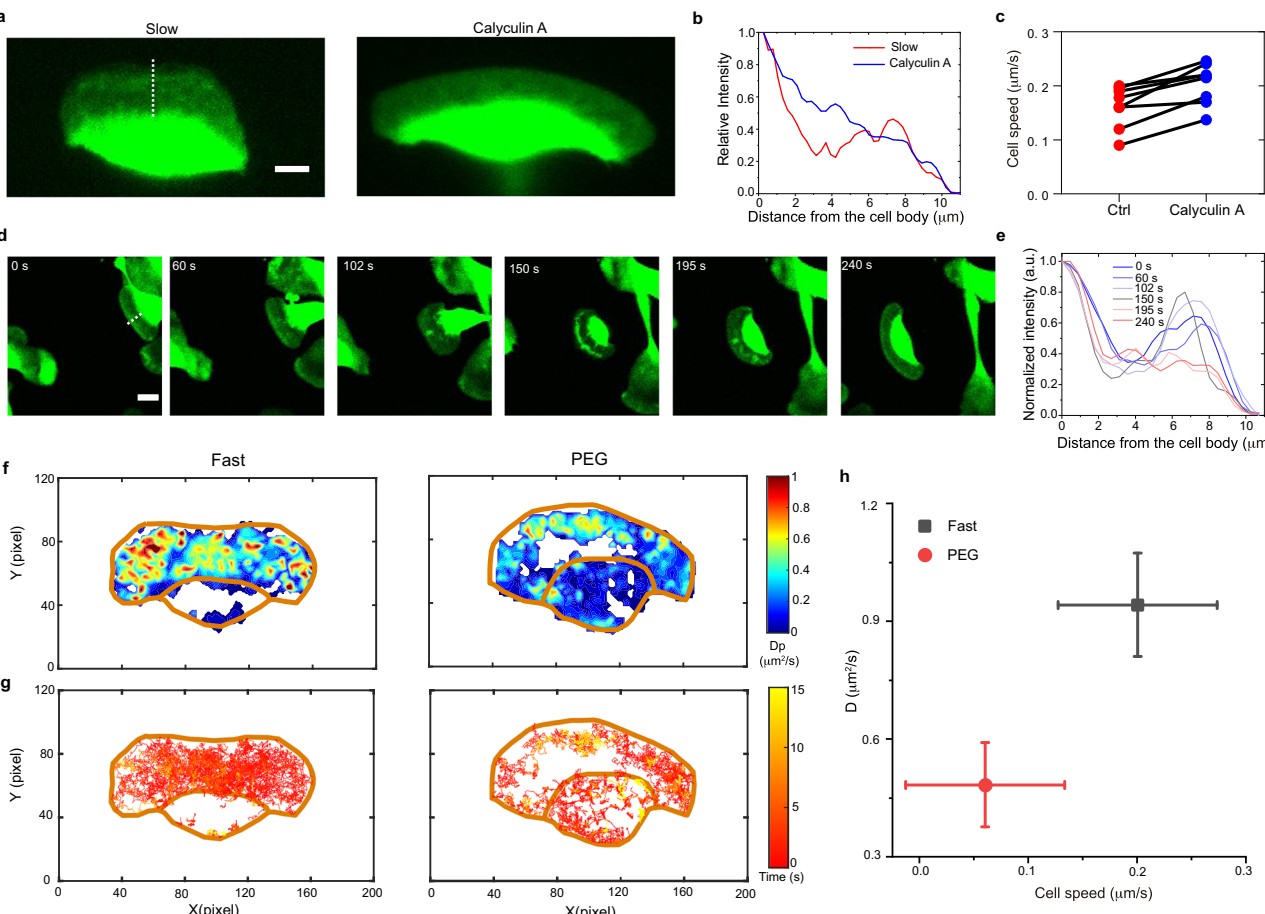

**Fig. 3 | Reversible transitions in both the lamellipodium structure and intracellular dynamics take place when regulating cell migration speed. a** The CMFDA images of a live cell. For the cell in the slow mode with a fluorescence gap in the lamellipodium (left panel), 25 nM Calyculin A is added to enhance the myosin II activity. The cell speed then increases, and the gap disappears accordingly (right panel). Scale bar, 10 μm. **b** Intensity profiles of CMFDA along the white line across the lamellipodium from the cell body to its leading edge, as shown in (**d**). **c** Comparison of cell speed of the same cell before and after the calyculin A treatment (n = 7). **d** The lamellipodium reversibly changes from the gap state to the flat shape when the cell separates from other cells and loses mechanical loading.

**e** The intensity profile of CMFDA along the white line from the cell body to the leading edge shows the dynamical changes of lamellipodium fluorescence, which indicates the lamellipodium thickness. In the beginning, the CMFDA intensity at the rear lamellipodium is obviously lower than that at the front part. After 195 s, when the cell is separated from the population, the intensity difference between the rear and front lamellipodia is eliminated. **f, g** Diffusion maps (**f**), and projection of trajectories (**g**) of QDs for the same cell before and after the treatment with 1% PEG. **h** Average D and migration speeds for cells before and after PEG treatment (fast, n = 6; PEG, n = 6). Data are shown as mean ± SD. Source data are provided as a Source Data file.

thickness of keratocytes that is below the optical diffraction limit, we apply 3D SPT with the axial resolution at about 35 nm, which is over ten times better than that of conventional 3D imaging methods. Based on the 2D SPT microscopy, the 3D SPT additionally captures a defocused plane, in which a single QD particle appears as a bright spot surrounded by a diffraction ring[52–55] (Fig. 4a). The ring radius increases linearly with the axial distance of the particle away from the cover glass (Fig. S19). A narrow distribution of diffraction radius of the diffusing QDs corresponds to a thin lamellipodium, whereas a wide one corresponds to a thick lamellipodium (Fig. 4b, c). In normal fast-migrating cells, the lamellipodium thickness is measured to be 234 ± 79 nm (Fig. 4d, e), which is consistent with previous results measured by transmission electron microscopy (TEM)[49] and atomic force microscope (AFM)[50]. In slow cells, the thickness of the rear lamellipodium is too small and cannot be measured because QDs are excluded, whereas that of the swollen-up-front lamellipodium is as much as 556 ± 170 nm.

To further validate the measured thickness of lamellipodia, we performed AFM on fixed cells, where the measured heights of lamellipodia should be no greater than their thicknesses considering the spreading feature of keratocytes (Fig. 4h–j). As expected, the measurements are consistent with the results for living cells by using 3D SPT, with the lamellipodium heights measured by AFM only being slightly smaller due to the methodological difference and cell fixation. More importantly, a clear comparison of the heights between rear and front lamellipodia in slow cells is revealed by AFM, with the rear lamellipodium measured to be 76 ± 33 nm in height. We notice that the local thickness at the rear lamellipodium is approaching the size of QDs, supporting our speculation that the QDs are excluded from the rear area due to the locally reduced thickness.

While much work has studied lamellipodia and their roles in cell migration, they mainly focused on the 2D structures and dynamics of lamellipodia with light microscopy[11,14–16,21,27,56]. However, since the lamellipodium thickness is close to the optical diffraction limit, the

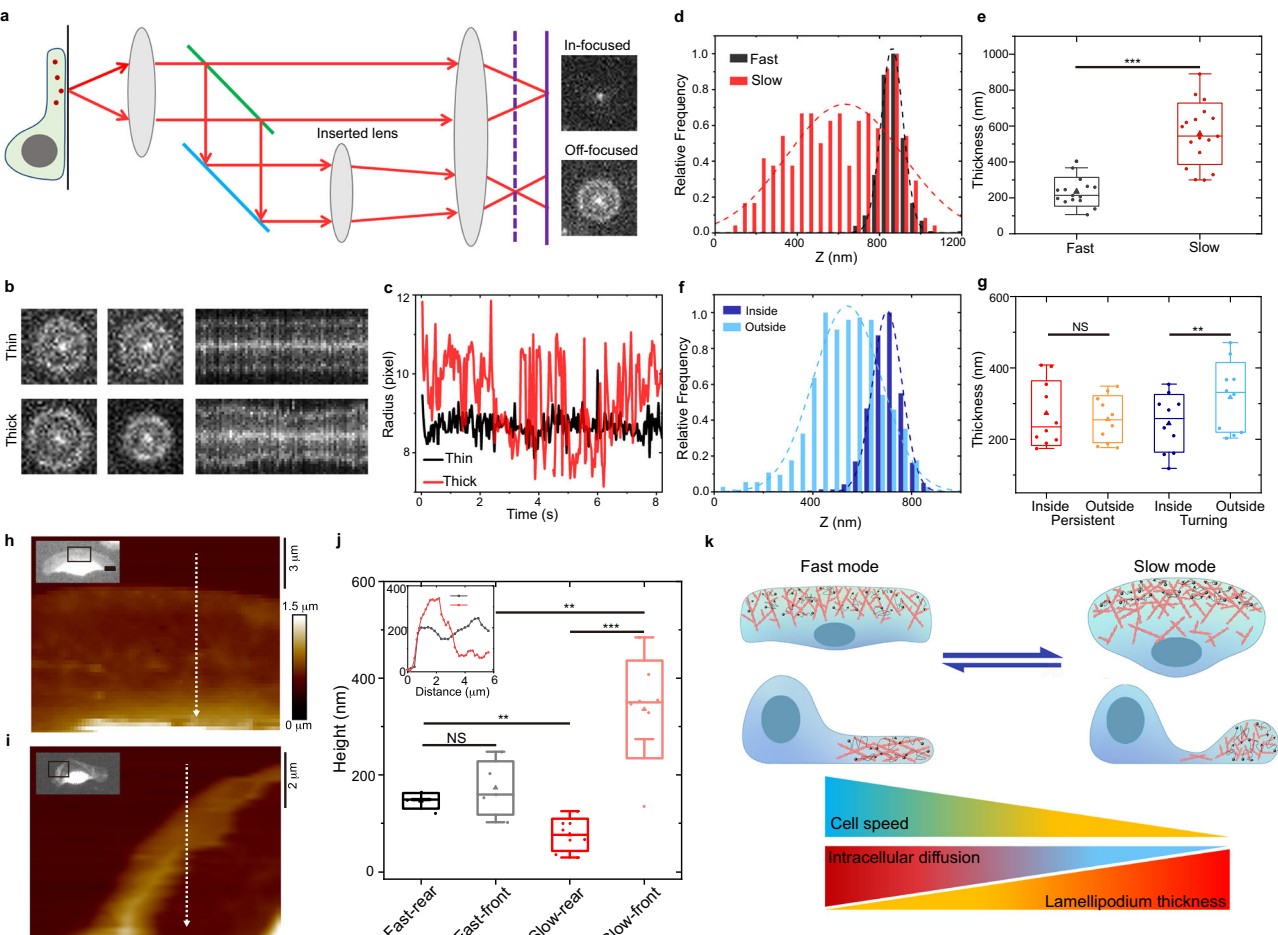

**Fig. 4 | Measurements of lamellipodium thickness and a proposed mechanism for cell migration. a–c** A schematic of the 3D SPT setup. The axial position of QD is determined by its diffraction ring radius from the off-focused channel (**a**). A thick lamellipodium has a larger fluctuation of diffraction ring radius of diffusing QDs (**b**, **c**). **d** Distributions of diffraction ring radius of QDs in fast and slow cells. **e** The measured lamellipodium thicknesses of fast (*n* = 15) and slow (*n* = 17) cells (*p* value = 2.1E−7). **f** Distributions of diffraction ring radius of QDs in the inside and outside of lamellipodium of a turning cell. **g** The measured lamellipodium thicknesses at the inside and outside parts of persistent (*n* = 10) and turning (*n* = 10) cells. Two-sample pair *t*-test; *p* = 0.32, the comparison of inside and outside for persistent cells; *p* = 0.0038, turning cells. **h, i** Typical AFM images of fast (**h**) and slow (**i**) moving cells. The inset images are the corresponding CMFDA images of the cells. The black boxes on the inset images indicate the region measured by AFM

experiments. **j** Statistics of height in the front and the rear parts of lamellipodium (fast, *n* = 5; slow, *n* = 8). *p* = 0.35, 0.0011, 0.0076, 7.6E−5 for fast-rear and fast-front, fast-rear and slow-rear, fast-front and slow-front, slow-rear and slow-front, respectively. Inset: Height profile along the dashed line from the lamellipodium edges to the inside of the lamellipodium (along the arrow direction). **k** Proposed mechanism for dynamic switching of a keratocyte cell between the fast and the slow migration modes. The cell in each mode is presented in both top and side views, with the actin filaments inside the lamellipodium in orange. Black dots and lines with arrows represent the QDs and their trajectories. All statistical tests are two-sided. **P < 0.01; ***P < 0.001; NS not significant. All the boxes represent SD, where the center lines represent the medians, whiskers represent 1.5× outliers, and triangles represent the averages. Source data are provided as a Source Data file.

measurement of thickness profiles of lamellipodia is challenging. Even though the TEM could measure the lamellipodium thickness[49], it is restricted to the fixed cells and cannot reveal the dynamical changes with cell migration behaviors. Here, by determining the thickness profile of lamellipodia in migrating keratocytes at nanometer resolution with 3D SPT, we find the swollen-up front and thinned-down rear lamellipodia in slow cells, which is distinct from the bending[57] or ruffling[58] at the leading edges of lamellipodia. Moreover, with the 3D SPT, the positive correlation between the cell migration speed and 3D intracellular diffusion rate still exists (Fig. S20), excluding the impact of axial diffusion on our results.

Collectively, we have shown that when cells are in the fast migration mode, they will have a flat lamellipodium, a fast intracellular diffusion, and uniformly distributed molecule components in the lamellipodium. In contrast, when cells are in the slow migration mode, the cells have an uneven lamellipodium with a swollen-up front and thinned-down rear, reduced intracellular diffusion, and compartmentalized macromolecules in the lamellipodium. More importantly, we demonstrate that the migration mode switching is reversible under different conditions, and corresponding transitions in both the lamellipodium structure and intracellular dynamics take place concomitantly (Fig. 4k).

### Left-right symmetry is broken in the lamellipodia of turning cells

In addition to changing migration speed, making turns is another typical migratory behavior of keratocyte cells[59]. To investigate if intracellular diffusion remains correlated with cell migration when they turn, we study the keratocyte cells that make turns. Initially, a cell is migrating persistently, in which the diffusion map of QDs is uniform (Fig. 5a, b). Interestingly, however, when the cell approaches another cell, the intracellular diffusion inside the cell becomes left-right asymmetric (Fig. 5c, d), with the average diffusion rate in the inside part of the lamellipodium being 40% higher than that in the outside part. This asymmetry is further illustrated by the MSD curves of QDs in both persistently migrating cells and turning cells (Fig. 5e, f). We also examine the lamellipodium thickness using CMFDA and observe that the inside part has lower fluorescence intensity than the outside part (Fig. 5g and Fig. S21). We then use 3D SPT to measure the lamellipodium thicknesses more directly and find that the thicknesses are $245 \pm 80$ nm in the inside part and $318 \pm 97$ nm in the outside part of the turning cells, while they are similar on both sides ($273 \pm 91$ and $256 \pm 66$ nm) for persistently migrating cells (Fig. 4f, g).

The above results reveal that the left-right symmetry of the lamellipodium structure and the intracellular diffusion has been broken in turning cells. Similarly, as in slow-migrating cells, the lamellipodium region with increased thickness in turning cells corresponds to a lower intracellular diffusion rate.

Moreover, from the above studies, we hypothesize that the molecular crowding change is perhaps the major factor in modulating intracellular diffusion when cells switch between the fast and the slow migration modes. The turning of cells inspires us to elaborately tune the molecular crowding only on one side of the lamellipodium and then examine its effect in regulating cell migration behaviors. By gently injecting PEG via a microneedle, the increased osmotic pressure is applied locally to a directly migrating cell (Fig. 5h), in which the hypertonic region is visualized by mixing a cell-impermeant tracer with 1% PEG (Fig. 5i). Interestingly when one side of the lamellipodium of the cell is entering the hypertonic region, the cell makes a turn towards the other side (Fig. 5j), with a local deformation of the lamellipodium where PEG is applied (Fig. 5k). Considering the correlation between intracellular diffusion rate and osmotic pressure we have observed above (Fig. 3f–h), it is reasonable to think that the locally increased osmotic pressure at the first side of the lamellipodium generates a left-right asymmetric intracellular diffusion. Consistent with the

spontaneously turning cell (Fig. 5a–d), the cell makes a turn towards the direction with relatively higher intracellular diffusion. The experiments further disclose the crucial role of the coupling of intracellular diffusion and lamellipodium structures in cell migration.

## Discussion

In this work, from the perspective of intracellular diffusion, we reveal that the intracellular diffusion variation and lamellipodium structure deformation are coordinated and reversible while the keratocyte cells switch their migration behaviors. More importantly, we find a new migration mode with slow speed, the feature of which includes a swollen-up front and thinned-down rear lamellipodium, decreased intracellular diffusion, and compartmentalized macromolecule distribution in the lamellipodium (Fig. 4k).

The deformation of 3D lamellipodium is consistent with the reorganization of actin networks in slow cells, in which a denser actin network in front lamellipodia was shown by STED imaging (Fig. 2d) and by electron microscopy recently[18]. It is reasonable to think that the actin network plays a role in the regulation of lamellipodium shape here[3,8,32]. Moreover, within the front lamellipodium of slow cells, the increase of actin density, together with the compartmentalization and concentration of large molecules, will increase the degree of molecular crowding locally and hence reduce the intracellular diffusion rates[31,40,41], considering that the cell volume remains similar (Fig. S9). To understand the coordination of 3D lamellipodium structure and intracellular diffusion with the cell migration behaviors, we propose a mechanism that involves the front-localized actin polymerization and increased molecular crowding in the lamellipodium. It has been revealed that actin filaments elongate only when their tips contact the cell membrane at the leading edge, or they will be capped, and the elongation will then be terminated when detaching from the membrane[3,18]. Therefore, when a cell is in the normal fast migration mode, only actin filaments almost perpendicular to the membrane would polymerize fast enough to maintain contact with the advancing membrane. However, when the membrane forward motion is reduced, either happening spontaneously or being induced by drugs or dragging forces, more actin filaments with a broader range of angles can maintain contact with the slowed membrane and polymerize continuously, which would form a denser actin network in the front lamellipodium of slow cells. The front part of the lamellipodium with the denser actin network swells up, and the rear part thins down accordingly as the total cell volume remains invariant. As a result, the large molecules are excluded from the rear part and compartmentalized in the swollen-up-front lamellipodium, which, together with the denser actin network, results in an increase of local crowding in the front lamellipodium. Accordingly, the intracellular diffusion would be significantly attenuated. Conversely, when the forward speed of the cell membrane is increased again, either happening spontaneously or being induced, only part of actin filaments perpendicular to the membrane could maintain contact with the leading membrane and continue polymerization. As such, the actin filament density gradually decreases in the front lamellipodium, which reduces the local molecular crowding and thus releases the cytosol back to the rear region. Finally, the lamellipodium returns to the flat shape, the intracellular diffusion becomes uniform, and the cell switches from the slow migration mode back to the normal fast migration mode (Fig. 4). Similarly, in turning cells, the speeds between the two sides of lamellipodium are imbalanced that the outside of the lamellipodium undergoes shrinkage while the inside expands (Fig. 5a, c). Due to the decreased forward speed of the membrane on the outside, the actin network becomes denser, and the local molecular crowding becomes stronger, which would increase the thickness of the outside of the lamellipodium and decrease the local intracellular diffusion rate.

The intracellular fluid flow is also an important factor in cell migration[22,35]. To test this, we study the migration of lamellipodial

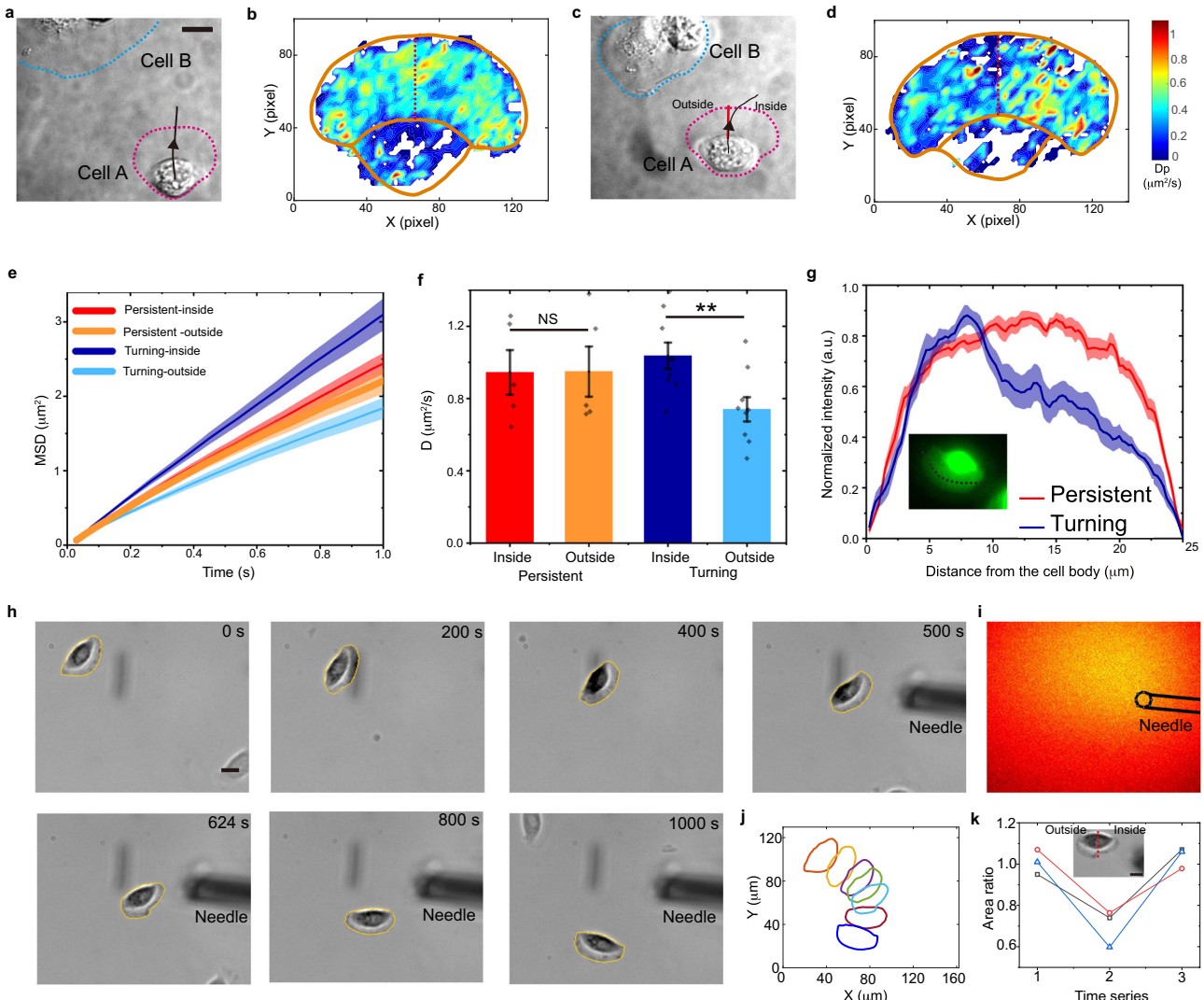

**Fig. 5 | Turning cells show left-right asymmetries in both intracellular diffusion and lamellipodium structure. a–d** A cell (denoted as Cell A) migrates upwards and then makes a turn to the right (**c**). The intracellular diffusion map in cell A changes from a normally left–right symmetric pattern (**b**) to an asymmetric one (**d**). Scale bar, 10 μm. **e** Comparison of MSD curves of diffusing QDs in the inside and outside halves of lamellipodium in persistently migrating cells (*n* = 5) and in turning cells (*n* = 9). The solid lines and the shaded regions represent mean ± SE. **f** Comparison of the QD diffusion rates. Error bars indicate the SEM. All statistical tests are two-sided. **, *p* < 0.01, with *p* = 0.0087; NS not significant, with *p* = 0.98. **g** Relative intensity of CMFDA in the lamellipodia of persistently migrating (*n* = 12) and turning cells

(*n* = 11) along the cell outline from outside to inside, as indicated by the white line in the CMFDA image of a turning cell (insert). The solid lines and the shaded regions represent mean ± SE. **h** A cell changes its direction in response to the left–right asymmetric osmotic pressure. A glass microneedle is placed near the cell and flows 1% PEG to increase the local osmotic pressure. Scale bar, 10 μm. **i** The hypertonic region is indicated by the cell-impermeant Fluo-4 pentapotassium salt that is mixed with PEG. **j** Locations and orientations of the cell at different times. **k** The area ratio of inside and outside parts of lamellipodium is plotted as a function of the time index (before (1), during (2), and after (3) turning) for three cells. Scale bar, 10 μm. Source data are provided as a Source Data file.

fragments, in which the intracellular fluid flow is changed due to the absence of myosin contractions at the cell rear[34]. We find that the treatment of Bleb has no obvious effect on fragment speed and properties, which is consistent with previous results[60]. However, when treated by LatA, the fragments are slowed down, and more importantly, both lamellipodium deformation with swollen-up front and thinned-down rear, and the compartmentalized macromolecule diffusion are observed (Fig. S22), similarly as in the case of keratocyte cells in the slow mode. This experiment has eliminated the roles of intracellular fluid flow in intracellular diffusion and in the regulation of cell migration modes here. Moreover, we note that the mechanism for switching cell migration modes cannot be limited to the front-localized actin polymerization, but the possible role of membrane tension should also be considered[61,62], as it associates with the cell volume, cytoskeleton forces, and mechanical loading in our experiments.

Besides other biophysical mechanisms, such as the mismatch between actin polymerization and treadmilling[3,63], possible roles of molecular players aside from the actin[16,32,64], or hydrodynamic forces inside the cells[23,65] may also be involved in the lamellipodium structure and intracellular diffusion during the migration switch. This remains to be resolved in the future.

With the 3D SPT, we demonstrate the lamellipodium structure deformation in migrating cells and measure the lamellipodium thickness profile at nanometer resolution, highlighting the important roles of 3D lamellipodium structure in cell migration. More importantly, the coupling between the intracellular diffusion dynamics and lamellipodium structure improves our holistic understanding of cell motility from a physical perspective. In the thin flat lamellipodia of normal fast cells, the diffusion is mainly confined in the quasi-2D flat lamellipodium, providing a more efficient way for the translocation and

recycling of actin subunits and other relevant proteins and hence could promote cell migration[34,35]. In contrast, for cells in the slow migration mode, the macromolecules are mostly limited and get crowded in the front swollen part of the lamellipodium. Then the crowded conditions significantly attenuate the diffusion/translocation of these macromolecules[40,41], which may result in reduced cell migration. The 3D lamellipodium structure regulates intracellular diffusion, and in return, the coordinated diffusion also provides positive feedback to support the structural reorganizations of the lamellipodium.

The compartmentalization of macromolecules within the lamellipodia of slow cells represents a new kind of phase separation[30] to organize macromolecules in living cells through the transition of subcellular structure from quasi-2D to 3D spaces. Moreover, the systems-level multiscale understanding of cell migration paves the way to controlling cell migration behaviors and developing clinical therapies for related diseases.

## Methods

### Cell culture and reagents
Keratocytes were isolated from the scales of the AB zebrafish from the China Zebrafish Recourse Center. Briefly, without sacrificing the fish, one or two scales were taken and placed in peri-dishes with Leibovitz's L-15 medium (Gibco) supplemented with 10% fetal bovine serum (Gibco) and 1% antibiotic-antimycotic (Gibco). The scales were kept at 28 °C for approximately 24 h to allow the keratocytes to spread from the scales to the dish. To obtain single isolated cells, keratocytes in cell sheets were disaggregated by incubation for approximately 1 min in 0.25% trypsin and 2.21 mM EGTA (trypsin-EGTA). Lamellipodial fragments were obtained by treating cells with 100 nM staurosporine in 1 mL medium for 30 min at 35 °C[34,60]. To label the cell volume, 5 μM CMFDA (CellTracker, Invitrogen) was added to the cells in serum-free L-15 medium for 45 min, and then the cells were recovered over 30 min before QD or dextran loading. QD streptavidin conjugate 655 nm with PEG coating (Q10123MP) was purchased from Invitrogen. The QD size was determined by using the dynamic light scattering (Zetasizer Pro, Malvern) equipped with a narrow band filter to avoid probe fluorescence from three repeat measurements at 25 °C. 500-kD FITC-dextran (D7136) and 70-kD rhodamine B-dextran (D1841) were from Molecular Probes. Blebbistatin (ab120425) and Calyculin A (ab141784) were from Abcam. Latrunculin A and Jasplakinolide were from Sigma-Aldrich. Other chemicals were from Sigma-Aldrich unless otherwise stated.

### Probe internalization in living cells
The QDs and dextrans to be tracked were loaded into live cells via a pinocytic process[39,46]. The QDs were mixed with a hypertonic solution (Influx-pinocytic cell-loading reagent (I-14402, Invitrogen)) at concentrations of 20 and 5 nM for 2D and 3D particle trackings, respectively. For dextrans, the final concentration in the hypertonic solution is 0.1 mg/ml. Cells were incubated for 15 min at 28 °C in the solution, allowing the material to be carried into the cells via pinocytic vesicles. The cells were transferred to a hypotonic medium for 100 s, which resulted in the release of trapped QDs from the pinocytic vesicles within the cells. The cells were then left in a complete L-15 medium at 28 °C to recover for at least 30 min. Because the keratocytes are cells isolated from fish scales and the lamellipodia are very thin, we had to do 2 cycles of this loading process to ensure that we had enough QDs in the cells (at least 10 QDs for 2D tracking, 3 QDs for 3D tracking). Serum- and phenol-red-free L-15 medium was used for imaging.

### Imaging
Wide-field and highly inclined and laminated optical sheet (HILO) imaging was performed using a fluorescence microscope (IX73, Olympus) with a 60× oil TIRF objective (1.45 N.A., Olympus) and a back-illuminated EMCCD camera (DU-897 Ultra, Andor Technology) to image QD and the CMFDA. It was also equipped with an environmental chamber to provide optimal ambient conditions (28 °C) for cells during imaging. DIC optics (Olympus) was used while taking bright field images. Fluorescence images were acquired by an inverted optical microscope with two lasers: a 561-nm laser (Sapphire 561, Coherent) for the QDs (ET655/15 M) and Alexa 594 phalloidin, and a 488-nm laser for the CMFDA. In the emission path, fluorescence emission was collected through the objective and separated from the excitation lasers using a triple-band dichroic mirror and a triple-band emit filter. To reach two-color imaging, a dual-channel simultaneous-imaging system (DV2, 565, Photometrics) was used to image the QDs and the CMFDA. The two subimages were calibrated by imaging fluorescent beads (Invitrogen) that have an emission spectrum covering the two spectral windows. In 3D particle tracking, the dual view cassette was equipped with an amplitude beam splitter to separate the light from the microscope into two beams in a 3:7 intensity ratio. A defocusing lens (focal length = 300 mm) was added to the high-intensity beam path. The focused beam and the defocused beam were sent to one-half of the CCD camera. For continuous imaging, we obtained images with focus points in the right half part and diffraction rings in the left half part. The different lasers were combined and focused at the back focal plane of the objective. The on-off states of these lasers were controlled by mechanical shutters (Uniblitz LS6T2, Vincent Associates). The movements of QDs in the cells were acquired as movie files at 33 Hz for a total of 2000 frames. A confocal laser scanning microscope (Leica SP8 systems with a 40x oil-immersed objective of 1.3 N.A.) was also used. For STED imaging, the cells stained with SiR-actin were imaged under the STEDYCON (Abberior Instruments) and deconvolved using Huygens, with a lateral resolution at 35 nm.

### Cell boundary and speed
The plotting of cell boundaries was realized by ImageJ. Because the keratocytes showed almost no movement in 10 s, the maximum intensity Z-projection of the first and last 200 frames of QD images resulted in the appearance of outlines of cells. Then, we manually painted the boundaries with the Freehand selections tool in ImageJ. The boundaries were modified by selections of 'Fit spline' and 'interpolate' before being saved as XY coordinates. The cell speed was defined as the distance between the centroid of the first and last cell body boundaries divided by the time of the video. All the 2D trajectories translated to the last frame to remove the drift from the cell movement were used to calculate dynamic parameters such as the MSD, the diffusion rate, and the exponent α, and plot the diffusion map. To transform from the lab frame to the cell frame of reference[22,32], the whole-cell movement observed by CMFDA labeling was regarded as rigid motion, in which the changes in cell position as well as the angle between the long axis of the cell outlines were determined between consecutive frames. Then the contributions of cell translation and rotation movements were removed from the QD trajectories.

### Single-particle tracking
Analysis of the acquired image series was performed by using the ImageJ plugin Particle Tracker. To obtain enough trajectories in a cell, only those cells with over 10 QDs were analyzed in 2D tracking experiments. For each frame, individual QDs were detected and localized by adjusting parameters for radius, cut-off, and percentile. The percentile parameter was adjusted to capture the greatest number of QDs that were clearly visible. The cut-off parameter was set to exclude the possible few aggregations as well as the blurred QDs that were partially out of focus. The parameters of linking range and displacement were adjusted to link the detected particles between frames. In the case of SPT in cells, the linking range was set to 3 to bridge over short QD blinking events (no more than 3 frames). The localization

accuracy of the system was approximately 30 nm, as was determined from $MSD = 2\delta^2$ in each direction by imaging the immobilized QDs under the same imaging conditions (i.e., temperature, culture medium, $CO_2$) as those for cells, where $\delta$ is the localization accuracy[53].

## Fluorescent recovery after photobleaching

FRAP experiment was done on a Leica SP8 laser scanning confocal microscope with a 40× oil-immersed objective of 1.3 N.A. We used bi-directional scanning mode to image a 256 × 256 pixel region at 0.188 s per frame. The CMFDA was excited with a 488 nm laser at 0.5–1% power. Photobleaching was done after 3 frames. A region of the lamellipodium with an area less than 5% of the whole cell area was bleached by scanning the laser beam over the region at 100% laser power for 3–5 frames. We continued imaging 30 frames after bleaching. Then we used the polygon selections tool in Fiji to define the region of interest (ROI). For the ROI, we got the fluorescence intensity curve with the plugin Create Spectrum jru v1. The maximum and minimum values of the intensity curve are normalized to 1 and 0. The recovery rate is determined by fitting the normalized recovering curve to an exponential function, $I(t) = (1 - \varepsilon) - \alpha \cdot \exp(-t/\tau)$, where $\tau$ is the time for recovery, $\alpha$ is the fraction recovered, and $\varepsilon$ is the fraction of the ration that does not recover.

## Micromanipulation

Microneedles were produced from 1.0 mm diameter borosilicate glass by P-97 micropipette pullers (Sutter Instrument). The microneedle's position was controlled by a micromanipulator (InjectMan 4, Eppendorf). To apply mechanical loading to the cells, we applied the microneedle to tether the cell body. To generate a local hypertonic region near the cell, the microneedle was placed near the cell and flowed 1% PEG to increase the local osmotic pressure. The hypertonic region was indicated by the cell-impermeant Fluo-4 pentapotassium salt (F14200, Invitrogen) that was mixed with PEG.

## AFM measurement

We performed the AFM measurement with BioScope Resolve system (Bruker, Billerica, MA, USA). A silicon nitride probe with a tip radius of 70 nm was used (PFQNM-LC-A-CAL, Bruker). The operation mode is PeakForce QNM in Fluid, and the pre-calibrated spring constant is 0.075 N/m. The AFM image was measured through NanoScope Analysis software. Briefly, the keratocytes were labeled by CellTracker CMFDA and then fixed in 4% formaldehyde. We first performed epifluorescence imaging of the cell to obtain the lamellipodium structure and cell migration mode. Then we performed AFM imaging of the cell, with a resolution at 64 × 64 and a line scan rate of 0.2 Hz. After flattening the AFM image of the cell by using the plane fit tool, the height profile of the cell lamellipodium was measured by using the section tool in the analysis software.

## 3D single-particle tracking

3D single-particle tracking was performed using ImageJ and a user-defined program in MATLAB. To avoid the overlapping of diffraction rings from different QDs in a cell, only those cells with 3–5 QDs were analyzed in 3D SPT tracking. First, regions of interest were cut from the images with the focus part and the corresponding defocus part. The defocus part would be saved as image sequences. The two-dimensional (2D) trajectories were detected from the focus images. Second, because there was the transformational and rotational drift of the two parts of the images, the positions of the 2D trajectories were transformed to the defocus images with a transformational matrix. The transformational matrix was calculated from three pairs of immobile points chosen from the focus and defocus part images. Third, the regions at the center of the positions that transformed from the 2D

trajectories were cut to images of suitable sizes. The sizes covered the diffraction rings and avoided overlap with other rings as much as possible. Fourth, all images of rings were checked by eye to ensure that the fitting program worked properly and then fed through a bandpass filter to enhance contrast. Finally, each defocused image was fitted with a function of the form $Z = P_0 + P_1\exp[-P_2((x - x_0)^2 + (y - y_0)^2)] + P_3\exp[-P_4(((x - x_0)^2 + (y - y_0)^2)^{1/2} - R_0)^2]$, which is essentially a gaussian peak surrounded by a circle of radius $R_0$. The magnitude of $R_0$ varied depending on the $z$-displacement from the focal plane. The position $(x_0, y_0)$ is the center of the ring. The maximum detection range in the $z$ direction is up to 2 μm, which covers the thickness of the keratocyte lamellipodium that is less than 1 μm. Moreover, the different refractive indexes between the cell and water have been proven to have no obvious influence on the detection of $z$ displacement in our platform[55]. The axial resolution was approximately 35 nm, as was determined by calculating the axial MSD of immobilized QDs on glasses with a similar method as that in 2D.

## Data analysis

All data analyses were performed using a user-defined program in MATLAB. The trajectories were all filtered twice before the MSD analysis: (1) an intensity filter was used to remove the potential QD aggregates whose intensities were beyond 2 times that of the mean intensity of all the QDs, and the potential out-of-focus QDs whose intensities were below 0.05 times that of the mean intensity; (2) a fixation filter was used to remove the fixed QDs with displacements <100 nm.

Local MSD analysis was used to select and analyze the diffusion motion of the QD trajectories. For each point along a trajectory, a segment of 20 consecutive points was used to calculate the local MSD: $MSD (\tau) = <|\mathbf{r}(t + \tau) \cdot \mathbf{r}(t)|^2 >$, where $t$ is the acquisition time. Then, the MSD was fitted by the power law $MSD(\tau) = A\tau^\alpha$. The exponent $\alpha$ represents the nonlinear relationship of MSD with time, which contains information about the local motion modes: $\alpha \approx 1$ is free Brownian motion (e.g., free diffusion), $\alpha < 1$ subdiffusion, and $\alpha > 1$ superdiffusion. Similarly, the diffusion rate $D$ of the point was determined by fitting the 3 initial points of the local MSD curve with $MSD (\tau) = 4D\tau + c$, which is a commonly used standard in SPT. This fit was repeated for each point along the trajectory, resulting in a time series for the parameters $\alpha$ and $D$. The mean $D$ for a single cell was the average of all the local values of the points in the cell. In the analysis of dextran trajectories, a segment of 6 consecutive points was used to calculate the local MSD.

In the case of global MSD analysis, the whole trajectory of one specific QD was used to generate the MSD plot, which was used to analyze the global exponent $\alpha$ of all the trajectories.

## Diffusion map plotting

The plotting was performed using a custom-written program in MATLAB. All the trajectories were translated to the cell reference frame. To discard the QDs on the glass that was immobile, we only selected segments of trajectories for which points with the criteria $\alpha > 0.5$ were equal to or longer than 5 frames. First, square grids in pixels were generated to divide the area covered by the cell. Each grid node served as a point in the diffusion map, and the grid size corresponded to the spatial resolution of the diffusion map. The point value was equal to its local diffusion rate $D_p$. Second, the parts of all the segments within a distance threshold to the grid node were selected. Third, the ensemble-averaged MSD curve was calculated using the parts of segments. Only when the first three points of the MSD curve were calculated using more than five samples for each point was the ensemble-averaged MSD curve valid. Fourth, the local diffusion rate $D_p$ was determined by linearly fitting the initial three points of the MSD curve. The final grid size and distance threshold were chosen as 3 pixels (800 nm) and 2.8 pixels (750 nm), respectively, by considering

the QD diffusion rate and the sample size of the experiments. Finally, the contour map of $D_p$ was plotted in MATLAB with a built-in smoothing process.

### Fixed cell staining

Fixed cell staining was performed according to the methods described previously[63]. Briefly, cells were fixed in 4% formaldehyde in PBS for 15 min, permeabilized with 0.5% Triton X-100 for 10 min, and blocked with PBS-BT (3% BSA, 0.1% Triton X-100, and 0.02% sodium azide) for 30 min. Then, the cells were incubated with 1 µM SiR-actin Kit (CY-SC001, Cytoskeleton) or 0.2 µM Alexa594 phalloidin (A12381, Invitrogen) for 60 min to label F-actin or with 0.3 µM Alexa594 deoxyribonuclease I conjugate (D12372, Invitrogen) for 30 min to label G-actin. To avoid the loss of G-actin, the labeling, and imaging of G-actin are performed immediately after the cell permeabilization.

### Image processing

All image processing was performed using the ImageJ software, including producing the movies, aligning and merging two-color images or stacks, manually selecting boundaries, cutting images, and measuring the intensity along marked lines.

### Statistics and reproducibility

For comparison, a two-sided Student's *t*-test was applied using the Origin software. In all cases, *$p < 0.05$; **$p < 0.01$; ***$p < 0.001$; NS, not significant. All the measurements were taken in more than three independent experiments.

### Reporting summary

Further information on research design is available in the Nature Portfolio Reporting Summary linked to this article.

## Data availability

Data supporting the findings of this study are provided with this paper in the Source Data file. Source data are provided in this paper.

## Code availability

MATLAB scripts used in this work are available at https://github.com/chaojiang1901/3D-particle-tracking-MATLAB.

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

## Acknowledgements

We thank National Center for Protein Sciences at Peking University, particularly Dr. Siying Qin, for technical help with AFM. We thank Dr. Qihui Fan, Dr. Ruipei Xie, and Yuhang Li for their technical help with micromanipulation. We thank Optofem Technology Limited for providing STEDYCON STED. We thank Dr. Cheng Zhang for the technical help with dynamic light scattering. This study is supported by the National Natural Science Foundation of China to H.L. (12122402, 12074043), P.Y.W. (11874415), X.C. (12135003).

## Author contributions

C.J., P.Y.W. and H.L. conceived the project. C.J. and H.Y.L. performed the experiments; C.J., X.X., S.X.D., W.L., D.G., F.Y., X.C., M.G., P.Y.W. and H.L. analyzed and interpreted the data; C.J., X.X., S.X.D. and H.L. wrote the paper; P.Y.W. and H.L. supervised the project. All authors edited and approved the paper.

## Competing interests

The authors declare no competing interests.
