## [Peer Review File · Nature Communications]

Switch of cell migration modes orchestrated by changes of three-dimensional lamellipodium structure and intracellular diffusionREVIEWER COMMENTS

Reviewer #1 (Remarks to the Author):

In this work, Jiang et al. have reported the observation of the different migration modes of cells through SPT of QDs, and AFM. A new slow migration mode is discovered, with deformed lamellipodium, swollen-up front, thinned-down rear, reduced QD diffusion, and compartmentalized macromolecule distribution. A strong positive correlation between intracellular diffusion and cell migration speed can be observed. The fast mode and slow mode can be switched either spontaneously or through chemical or mechanical stimulation. Interestingly, the authors further confirm that both lamellipodium structure and intracellular diffusion dynamics are also changed, with left-right symmetry breaking, when a migrating cell is turning directions. This work brings new tools for the study of cell migration, and the new discoveries on the sub-cellular level give a fresh insight into the behavior of cell migration. The topic per se is relevant and interesting to a wide readership.

Comments:

1. The size of QDs plays important role in the behavior of QDs inside the cell. This should be clearly stated, with EM image for the size of QD.
2. What are cell frame and lab frame referring to?
3. What is the proper density of QDs to ensure successful SPT analysis, without being diluted too sparse? How was the 30-nm localization accuracy calculated?
4. On Page 6, the authors mentioned 'We find that in a fast cell, the diffusion is essentially heterogeneous across its whole lamellipodium (Fig. 2a)...', but as shown in Figure 2a, the intracellular diffusion of fast cells is more uniform than that of the slow cells. Where does the conclusion of uneven diffusion come from ?
5. Please add dual-color image on the same cell in Fig. 2 c and d, so that the CMFDA fluorescence image and SiR-actin image can be directly compared.
6. In Fig.2c, according to the author's description, the uneven distribution of quantum dots in the slow cells is due to the uneven distribution of the thickness of the lamellipodium. The quantum dots are squeezed from the thinner part to the front layer of the lamellipodium, which the author claims is a mechanism dominated by molecular crowding. Is there any direct evidence to confirm this? For example, mechanical pressure can be applied to slow the migration of fast cells by squeezing part of the lamellipodium.
7. On Page 11, the authors mentioned "Interestingly, it is observed that the addition of 1% PEG in the medium that increases the osmotic pressure has made the cell switch from the fast to the slow modes, with the QD excluded region in the lamellipodium area emerged accordingly (Fig. 3f-h)." Due to the dependence of cell volume on external osmotic pressure, the authors add increasing concentrations of PEG 300 to the medium; PEG 300 does not penetrate the cell membrane and thus increases the external osmotic pressure. The external osmotic pressure must be matched by the internal osmotic pressure, which is controlled by the concentration of ions and small proteins. To compensate for the increase of external osmotic pressure, the cell must increase its internal osmolyte concentration either by ion influx or water efflux. Thus, the cell volume decreases with increasing external osmotic pressure. To quantify

- the degree of crowding within cells, I suggest the authors to further measure the change in cell volume.
8. The authors find a new migration mode with slow speed, the feature of which includes a swollen-up front and thinned-down rear lamellipodium, decreased intracellular diffusion, and compartmentalized macromolecule distribution in the lamellipodium. Here, viscosity is an important factor affecting molecular diffusion. Whether the cell viscosity change when the cell migration mode changes?
 9. In Fig. 4a, the QDs were imaged with 3D defocus. In this configuration, both the positive and negative defocus generate the same off-focused pattern, right? How was the 35-nm axial resolution measured?
 10. Besides the change on the lamellipodium side, is there any change on the cell nuclei size and height in Fig. 4?
 11. P15 “Interestingly, however, when the cell approaches another cell, the intracellular diffusion inside the cell becomes left-right asymmetric (Fig. 5c, d). We then use 3D SPT to measure the lamellipodium thicknesses more directly and find that the thicknesses are 245 nm in the inside part and 318 nm in the outside part (Fig. 4f, g and Fig. S15).” The cells in Fig.5 are in different states from those in Fig. 4. I suggest the authors to further measure the lamellipodium thicknesses of the cells in Fig 5a and Fig. 5c.
 12. It is recommended that both the data and the code are uploaded to a data deposit for open science.

Reviewer #2 (Remarks to the Author):

This manuscript of Jiang and colleagues aims to address an interesting and important biological question on the role of intracellular diffusion in the control of cytoskeletal dynamics and cell migration. The authors provide experimental data demonstrating that spontaneous changes in the intracellular diffusion correlate with the density of actin filaments in lamellipodia, and hence, the speed of keratocyte migration. By tracking quantum dots taken up by the cell, the authors mapped out the diffusion speed across the cell and found that the diffusion of the quantum dots is faster in lamellipodia and slower in the cell body. By using an elegant combination of imaging techniques and pharmacological manipulations, the authors demonstrated that the suppression of intracellular diffusion results in thicker and less protrusive lamellipodia, while fast migrating cells exhibit high diffusion rate. Finally, the authors modulated the local diffusion rate by applying PEG and showed that local decrease of intracellular diffusion rate causes cell turning. Although this idea that intracellular diffusion regulates cell migration is extremely interesting to the field of cell biology and the data presented in the manuscript are novel, the experimental evidences supporting the main statements of the manuscript are often limited and inconclusive. My specific concerns and comments are as follows.

1. Because several important conclusions were made based on the quantum dot tracking assay, it is essential to show that the quantum dots are indeed diffusing freely in the cytoplasm, but not trapped in the vesicles or transported as cargos.
2. According to the authors, the data presented in Fig. S3 show that the intracellular actin flow does not have a detectable effect on the rate of quantum dot diffusion. Considering similar size of the quantum

dots and pores in lamellipodia actin network, it is hard for me to believe that the quantum dots are not impacted by the flow. This concern is further supported by data presented in Fig. 2 of the manuscript, which show a significant decrease in the intracellular diffusion rate upon actin depolymerization with LatA. Comprehensive analysis of quantum dot diffusion under the experimental conditions that suppress the actin flow (simultaneous inhibition of actin polymerization and myosin contractility and/or in cells fixed with PFA) would strengthen the manuscript.

3. The data presented in Fig 2 demonstrate that thicker regions of keratocytes (cell body and swollen lamellipodia) are characterized by the reduced intracellular diffusion rate. Can such difference in the diffusion be attributed to the movement of quantum dots in Z-direction?

4. In Fig 3 & 5, the authors attempted to use PEG to alter intracellular diffusion by applying a solution with high osmotic pressure. Other than the change in the diffusion maps, are there any other accompanying changes would support the effect of PEG on the cytoplasm crowdedness and membrane tension? For example, in Fig. 5h, I would expect to see some quantifiable local deformation on side of the cell where PEG was applied, but the cell shape seemed to be unchanged. Additional experimental data are also required to rule out the effect of PEG on cell migration through the membrane tension.

Minor suggestions

1. Most diffusion maps were created using data from a single cell. Since fish keratocytes have a relatively stereotypical cell shape, it makes sense to align maps from several similarly shaped and sized cells, and then compute the average of all the maps. This would help readers understand the phenotype at the population level.

Reviewer #3 (Remarks to the Author):

In this paper, Jiang and colleagues study how fish keratocytes regulate their migration speed and direction by switching in a reversible manner their lamellipodium structure and associated intracellular diffusion. The authors identify that keratocytes can spontaneously switch from a fast to a slower mode of migration. The latter mode is associated with deformation of the lamellipodium from a flat configuration to a swollen front and thinned rear as evidenced by fluorescence microscopy, AFM and 3D single nanoparticle tracking. This migration switch is also correlated with a reduced intracellular diffusion and a compartmentalized macromolecular distribution within the lamellipodium as shown using single quantum dots tracking and actin distribution. Furthermore, using treatment with pharmacological drugs affecting cellular speed, or micromanipulation using microneedles and cellular situations restraining cell speed, the authors demonstrated this newly identified migration speed switch and correlated changes in lamellipodium structure and intracellular diffusion to be reversible. The authors also demonstrated that the migration switch is also involved during cell turning when it occurs only on one-half of the cell supporting a left-right symmetry breaking. Finally using solution to control

locally the osmotic pressure, the authors propose that a molecular-crowding mechanism could be controlling the migration switch.

Overall, my opinion about the findings described in this manuscript is extremely positive. The novelty of this manuscript resides in identifying a new migration mode associated with changes in 3D lamellipodium structure and intracellular space organization. This is particularly striking as it was identified in fish keratocytes, one of the most studied cell model for cell migration. A noteworthy result is the starting point finding of this study, describing a positive correlation between cell migration speed and the intracellular diffusion probed by single nanoparticle tracking within the lamellipodium. This finding drove the authors to determine what changes in the organization of lamellipodium 3D architecture and intracellular space could be the basis of such correlation. In comparison to previous studies from Theriot's lab linking keratocytes speed and 2D shape, the present study introduced the variation of lamellipodium thickness as a crucial parameter controlling cellular speed. The introduction of 3D lamellipodium architecture, parameter that has been overlooked in the case of very flat keratocytes provides to this study, novelty and significance to the field of cell migration.

Nevertheless, a regret about this manuscript is that molecular mechanism proposed to control the migration switch in this study is not yet convincing. Most evidences about a molecular-crowding mechanism rely on osmotic pressure manipulation known not only to affect molecular density within the lamellipodium, but also membrane tension, an important parameter also described as controlling speed and morphology of keratocytes (Keren lab and Theriot lab). Furthermore the authors do not provide alternative biophysical mechanisms (mismatch between actin polymerization and treadmilling) or molecular players aside from actin that could regulate the control of lamellipodium thickness and macromolecular compartmentalization. In that sense, I feel that it would be informative to determine whether the migration speed switch or variations of lamellipodium thickness and/or intracellular diffusion are also observable for cell fragments that lack their cell body. At least, this would definitively exclude the described role of intracellular fluid flow in Keren et al, Nature Cell Biology 2009.

A strength of this study relies on the multiplicity and soundness of experimental approaches used to define 3D lamellipodium architecture and intracellular diffusion properties. This said, I feel that a too strong emphasis is given on the sole intracellular diffusion properties. This certainly emerges from the fact it is an expertise from the authors' lab and it is not clear yet what is the molecular mechanism at play. But in my opinion, intracellular diffusion and compartmentalization within lamellipodium are a consequence of the 3D lamellipodium reorganization during migration switch.

In summary, the findings described in this manuscript deserve publication in Nature communications. Nevertheless in the current form, the presentation of the results in a sequential manner is not concise and produce overwhelming repetitions as for instance results obtained for intracellular diffusion with quantum dots nanoparticles or fluorescently-tagged macromolecules (dextrans) that are described in different parts of the manuscript. Strikingly many information about the nanoparticles are not presented up-front but through out as for example their size or their inert biochemical properties. Before being fully accepted for publication in Nature Communications, I recommend that an effort

should be done to simplify the presentation of the results starting with the title that is highly descriptive and not conclusive. In this case, the emphasis should be given to the identification of a migration switch with the associated changes on 3D lamellipodium shape and intracellular diffusion properties.

In the following are only minor remarks, advices or questions:

Concerning the AFM measurements, why the authors did not perform lamellipodium thickness measurements in live keratocytes and used fixed samples with all the risks associated?

Concerning G-actin labeling, I was wondering whether cell permeabilization could lead to loss of G-actin present in the cytosol and bias the ratio measurement between f-actin and g-actin?

In the introduction section, the sentence "Given that the active fluid flow exists, it cannot fully account for an intracellular transport to match the cell speed" is not easy to understand and need to be revised.

On page 6, in the sentence "in a fast cell, the diffusion is essentially heterogeneous across its whole lamellipodium", it is not clear what the authors mean by heterogenous. Do they mean continuous or uncompartimentalized?

In figure 2, images of CMFDA distribution in panel 2c appear to display a very low contrast with background signals from outside the cell to be as intense as in the lamellipodium. It would be worth matching the background signal outside the cell for all 4 images.

On page 8, in the sentence "the presence of strikingly excluded gap regions in the lamellipodia of slow cells is resulted from a physical barrier", "is resulted" should be replaced by "is resulting".

On page 11, in the sentence "After 195 s when the cell is isolated from the population", "isolated" should be replaced by "separated".

In the discussion (page 17), "reversable" should be replaced by "reversible"

We thank the reviewer for the careful reviewing of our manuscript and the very helpful comments and suggestions. Accordingly, we have addressed all comments and questions point by point below. For convenience, the reviewers' comments are shown *in black*, our responses are shown *in blue*, and the corresponding changes in the manuscript are shown *in red* in the revised manuscripts.

RESPONSE to Reviewer #1:

In this work, Jiang et al. have reported the observation of the different migration modes of cells through SPT of QDs, and AFM. A new slow migration mode is discovered, with deformed lamellipodium, swollen-up front, thinned-down rear, reduced QD diffusion, and compartmentalized macromolecule distribution. A strong positive correlation between intracellular diffusion and cell migration speed can be observed. The fast mode and slow mode can be switched either spontaneously or through chemical or mechanical stimulation. Interestingly, the authors further confirm that both lamellipodium structure and intracellular diffusion dynamics are also changed, with left-right symmetry breaking, when a migrating cell is turning directions. This work brings new tools for the study of cell migration, and the new discoveries on the sub-cellular level give a fresh insight into the behavior of cell migration. The topic per se is relevant and interesting to a wide readership.

RESPONSE: We thank the reviewer for the positive and encouraging comments!

Comments:

1. The size of QDs plays important role in the behavior of QDs inside the cell. This should be clearly stated, with EM image for the size of QD.

RESPONSE: We thank the reviewer for the good suggestion. To determine the size of QDs, we first tried to take the transmission electron microscope image of QDs, but found that only the core-shell nanoparticle of the QD can be imaged, without the polymer coating on the QD surface. Then we performed the dynamic light scattering (DLS) experiments to measure the hydrodynamic radius of QDs. The technique is commonly used to determine the size of particles in solution ranging from 1 to 1000 nm. The measured size of QD is $30 \text{ nm} \pm 0.7 \text{ nm}$ (please see the Figure R1 below), which is consistent with previous reports (Keren, Kinneret, et al., Nature cell biology, 11, 1219 (2009)).

Accordingly, we have clearly stated the size of QDs in the revised manuscript on page 4:

To investigate the intracellular diffusion dynamics of fish keratocytes, we use fluorescent quantum dots (QDs) coated with polyethylene glycol (PEG) as the nonspecific diffusing probes, with a size around 30 nm^{22} ,
39, 43.

We also added information of the DLS in the Methods section on page 20:

The QD size was determined by using the dynamic light scattering (Zetasizer Pro, Malvern) equipped with a narrow band filter to avoid probe fluorescence, from three repeat measurements at $25 \text{ }^\circ\text{C}$.

Figure R1. Distribution of measured QD sizes measured by DLS. The mean value is 30.0 ± 0.7 nm from three repeat measurements.

2. What are cell frame and lab frame referring to?

RESPONSE: We are sorry for not describing it clearly. The lab frame refers to the stationary frame of reference used in the laboratory, and the cell frame is a moving frame of reference that corresponds to the movement of a moving cell. In the lab frame reference, we essentially treat the cell as a rigid body. So that we could determine both the translation and rotation movements of the entire cell, by measuring the changes in cell position, as well as the angles between the long axis of the cell outlines, from frame to frame. The QD trajectories in the reference of lab frame are then transformed into the cell frame, by removing the cell translation and rotation movements.

Accordingly, we have cited reference papers for the cell frame in the manuscript on page 5, and added more description in the Methods section on page 22:

To transform from the lab frame to cell frame of reference^{22, 32}, the whole-cell movement observed by CMFDA labelling was regarded as rigid motion, in which the changes in cell position as well as the angle between the long axis of the cell outlines were determined between consecutive frames. Then the contributions of cell translation and rotation movements were removed from the QD trajectories.

3. What is the proper density of QDs to ensure successful SPT analysis, without being diluted too sparse? How was the 30-nm localization accuracy calculated?

RESPONSE: They are good questions. First, for the proper density of QDs to ensure the single-particle tracking (SPT) analysis in keratocyte cells, we usually image the cell with over 10 QDs to obtain enough trajectories in 2D SPT experiments. However, in 3D SPT experiments, we typically image the cell only with 3-5 QDs, which would avoid the overlapping of diffraction rings from different QDs in a cell. Second, the

localization accuracy was determined from the immobilized QDs on the glasses under the same imaging conditions (i.e., temperature, culture medium, CO₂ supply), as commonly used before (Jiang et al., Chin. Phys. Lett., 37 078701 (2020); Thompson, et al., PNAS, 107, 17864 (2010)). By analyzing the $MSD = 2\delta^2$ in each direction, the localization accuracy δ is obtained, which is laterally 30 nm and axially 35 nm.

To make this clear to the readers, we have added more details in the Single particle tracking section on page 22:

The localization accuracy of the system was approximately 30 nm, as was determined from $MSD = 2\delta^2$ in each direction by imaging the immobilized QDs under the same imaging conditions (i.e., temperature, culture medium, CO₂) as those for cells, where δ is the localization accuracy.

4. On Page 6, the authors mentioned ‘We find that in a fast cell, the diffusion is essentially heterogeneous across its whole lamellipodium (Fig. 2a)...’, but as shown in Figure 2a, the intracellular diffusion of fast cells is more uniform than that of the slow cells. Where does the conclusion of uneven diffusion come from?

RESPONSE: We thank the reviewer to point out this issue. In this sentence, we intended to describe that the environment inside the fast cell is heterogeneous compared to the generally homogeneous solution. Indeed, when compared with the slow cells where the intracellular diffusion is compartmentalized in the front lamellipodia, the intracellular diffusion of fast cells is more uniform.

To avoid confusions, we have revised this sentence on page 6.

We find that in a fast cell, the diffusion of QDs is present across the whole lamellipodium.

5. Please add dual-color image on the same cell in Fig. 2 c and d, so that the CMFDA fluorescence image and SiR-actin image can be directly compared.

RESPONSE: Thanks for the good suggestion. To obtain the clear actin structures in the lamellipodia, we obtain the super-resolution image of the SiR-actin under a commercial STED microscope, which is kindly provided by a company. But the STED microscope is not equipped with the excitation laser for the CMFDA, therefore, we cannot add the CMFDA image of the same cell. To directly compare the CMFDA and SiR-actin, we have provided the images of CMFDA and SiR-actin for the same cells, which are imaged under our confocal microscope. Moreover, we also provide the images of the CMFDA and G-actin for the same cells. It could be seen that the distribution of SiR-actin coincides with the CMFDA, and the G-actin also show similar distribution as the CMFDA.

Please find the added figures in the new Figure S12:

Fig. S12. a, Fluorescence images of CMFDA (upper panels) and F-actin (lower panels) for migrating cells with fast and slow speeds, and for cells treated with 10 nM latrunculin A (LatA), or 50 μ M blebbistatin (Bleb). **b**, Fluorescence images of CMFDA (upper panels) and G-actin (lower panels) under the same conditions as those in (a). Scale bar, 10 μ m.

6. In Fig.2c, according to the author's description, the uneven distribution of quantum dots in the slow cells is due to the uneven distribution of the thickness of the lamellipodium. The quantum dots are squeezed from the thinner part to the front layer of the lamellipodium, which the author claims is a mechanism dominated by molecular crowding. Is there any direct evidence to confirm this? For example, mechanical pressure can be applied to slow the migration of fast cells by squeezing part of the lamellipodium.

RESPONSE: We thank the reviewer for the excellent comment and suggestion. To explain the observed swollen lamellipodium in the front of slow cells, the mechanism of molecular crowding is proposed, by having eliminated other possible factors including the roles of intracellular fluid flow and nonspecific binding of probe. Indeed, we cannot directly regulate the molecular crowding only at the front lamellipodium to provide further evidences.

As suggested by the reviewer, we have tried to mechanically press down the lamellipodia of fast cells by using a microneedle. As the cell lamellipodium is quite thin (\sim 200 nm), we must vertically lower the microneedle towards the rear part of lamellipodium, and make it as close to the glass substrate as possible to effectively press the lamellipodium. However, we found that because the keratocyte moves very fast, the forward movement of the cell body would be inevitably blocked by the microneedle, before we readjust the microneedle (*please see the Figure R2 below*). Therefore, although this manipulation would obviously slow the cell, we cannot attribute it to the squeezing of the lamellipodium. Here, our microneedle is manipulated by a manual control. The microneedle cannot be controlled precisely and fast enough. We think if we could have

an automated programming-controlled handle, it would be better. Still, we would like to thank the reviewer's great suggestion.

Figure R2. The horizontal microneedle has blocked the cell body and hindered the cell migration.

To provide more clues to support that the quantum dots are squeezed from the rear part to the front part of lamellipodium, we analyze the dynamic change of intracellular QD distributions when a cell switches from fast to slow modes. As shown in the new Figure S10, it is observed that the QDs in a migrating cell show a uniform distribution across the lamellipodium at the beginning. From 10.5 s, an excluded area with no QDs gradually appears in the rear part of lamellipodium, and the QDs are compartmentalized within the front lamellipodium. Further analysis of the moving shows that such dynamic change of QD distribution is accomplished within 3 seconds. Considering that the above process is happened due to the changing of 3D structure of lamellipodium, the very fast process is more likely to be attributed to the physical squeezing by the lamellipodium, not for other biological reasons. Please find the results in the new Figure S10.

We revised the manuscript on page 8: Consistent with this, when the migrating cell slows down, we have observed the rapid translocation of intracellular QDs from the rear part of lamellipodium to the front part, taking place within ~3 s (Fig. S10).

Fig. S10. The QDs rapidly translocate from the rear to the front parts in the lamellipodium. **a**, Images of QDs in a migrating cell at different time points. The orange triangle points to the region where the QD intensity changes obviously. Scale bar, 10 μm . **b**, Kymograph of the QD intensity along the red line shown in (a). The colors from blue to red represent the increase in intensity. Black box indicates the time period from 10.5 to 13.5 s, during which the translocation of QDs from the rear to front parts in the lamellipodium is completed. **c**, Intensity profile along the red line in (a) is plotted as a function of distance from the rear to the leading edges of lamellipodium. Before 10.5 s, the intensity in the front and rear parts of lamellipodium is relatively equal. Between 10.5 and 13.5 s, the intensity decreases in the rear region while increases in the front region. After 13.5 s, the intensity in the front region is consistently higher than in the rear region, resulting in a pattern with a gap area in the rear part of the lamellipodium.

Moreover, we realize that we cannot exclude other factors such as membrane tension that may take part in. Therefore, we have added more discussions on other possible mechanisms in the Discussion section. And we rewritten the proposed mechanism involving the front-localized actin polymerization and increased molecular crowding in the lamellipodium, to explain how cells spatiotemporally coordinate the intracellular diffusion dynamics and the lamellipodium structure in regulating their migration behaviors. Please find the new paragraph in the Discussion section on page 17 and 18:

The deformation of 3D lamellipodium is consistent with the reorganization of actin networks in slow cells, in which a denser actin network in front lamellipodia was shown by STED imaging (Fig. 2d) and by the electron microscopy recently¹⁸. It is reasonable to think that the actin network plays a role in the regulation of lamellipodium shape here^{3, 8, 32}. Moreover, within the front lamellipodium of slow cells, the increase of actin density together with the compartmentalization and concentration of large molecules, will increase the degree of molecular crowding locally and hence reduce the intracellular diffusion rates^{31, 40, 41}, considering that the cell volume remains similar (Fig. S9). To understand the coordination of 3D lamellipodium structure and intracellular diffusion with the cell migration behaviours, we propose a mechanism that involves the front-localized actin polymerization and increased molecular crowding in the lamellipodium.

The intracellular fluid flow is also an important factor in cell migration^{22, 35}. To test this, we study the migration of lamellipodial fragments, in which the intracellular fluid flow is changed due to the absence of myosin contractions at the cell rear³⁴. We find that a treatment of Bleb has no obvious effect on fragment speed and properties, which is consistent previous results⁶⁰. However, when treated by LatA, the fragments are slowed down, and more importantly, both lamellipodium deformation with swollen-up front and thinned-down rear, and the compartmentalized macromolecule diffusion are observed (Fig. S22), similarly as in the case of keratocyte cells in the slow mode. This experiment has eliminated the roles of intracellular fluid flow in the intracellular diffusion and in the regulation of cell migration modes here. Moreover, we note that the mechanism for switching of cell migration modes cannot be limited to the front-localized actin polymerization, but the possible role of membrane tension should also be considered^{61, 62}, as it associates with the cell volume, cytoskeleton forces and mechanical loading in our experiments. Besides, other biophysical mechanisms such as the mismatch between actin polymerization and treadmilling^{3, 63}, possible roles of molecular players aside from the actin^{16, 32, 64}, or hydrodynamic forces inside the cells^{23, 65}, may also be involved in the lamellipodium structure and intracellular diffusion during the migration switch. This remains to be resolved in the future.

7. On Page 11, the authors mentioned “Interestingly, it is observed that the addition of 1% PEG in the medium that increases the osmotic pressure has made the cell switch from the fast to the slow modes, with the QD excluded region in the lamellipodium area emerged accordingly (Fig. 3f–h).” Due to the dependence of cell

volume on external osmotic pressure, the authors add increasing concentrations of PEG 300 to the medium; PEG 300 does not penetrate the cell membrane and thus increases the external osmotic pressure. The external osmotic pressure must be matched by the internal osmotic pressure, which is controlled by the concentration of ions and small proteins. To compensate for the increase of external osmotic pressure, the cell must increase its internal osmolyte concentration either by ion influx or water efflux. Thus, the cell volume decreases with increasing external osmotic pressure. To quantify the degree of crowding within cells, I suggest the authors to further measure the change in cell volume.

RESPONSE: This is an excellent suggestion. We have measured the volumes of both the cell and nuclei in control cells and cells treated with 1% PEG. Indeed, we observed a significant reduction in both cell and nuclei volumes in the PEG-treated cells. These findings suggest that the intracellular crowding should increase after the treatment, which further supports our argument that molecular crowding can influence the migration mode of keratocytes.

We have shown the results in the added Figure S9, and revised the manuscript on page 8:

Moreover, we find no differences in the cell volume, or in the nuclei volume and height between fast and slow cells (Fig. S9).

Fig. S9. Characterization of cell and nuclear volumes. **a, b,** The cell (**a**) and nuclear (**b**) volumes for fast- and slow-moving cells. They remain the same when cells change the migration modes (fast, $n = 19$; slow, $n = 16$). **c, d,** Comparison of the cell (**c**) and nuclear (**d**) volumes between control cells and those treated with 1%

PEG (ctrl, n = 35; 1% PEG, n = 39). With the 1% PEG treatment, noticeable decreases in both volumes were observed. **e, f**, The heights of cell body (**e**) and nuclei (**f**) for fast- and slow-moving cells, as well as for cells under 1% PEG treatments (n = 15). The boxes represent SD percentiles, the whiskers represent extremes in data, and triangles express averages. The cells were labeled with CMFDA and the nuclei with Hoechst33342. We performed 3D scanning imaging using a confocal laser microscopy. The volumes were measured by counting voxels in the z-stack images, with a 3D pixel size of 200 nm.

8. The authors find a new migration mode with slow speed, the feature of which includes a swollen-up front and thinned-down rear lamellipodium, decreased intracellular diffusion, and compartmentalized macromolecule distribution in the lamellipodium. Here, viscosity is an important factor affecting molecular diffusion. Whether the cell viscosity change when the cell migration mode changes?

RESPONSE: This is an excellent and insightful question. In the complex environment of cytoplasm, due to the active fluctuations and the molecular crowding, it is difficult to separate the influence of multiple factors on intracellular diffusion, and to measure the fluid-phase viscosity of the cytoplasm. Previous studies on the origin of reduced intracellular diffusion have shown that the fluid-phase viscosity of cytoplasm is not much greater than that of water (Dix and Verkman, *Annu. Rev. Biophys.*, 37, 247 (2008)), but the probe collisions with the cytoplasm crowders are the principal diffusive barrier for the reduced diffusion in cells. Therefore, the experienced viscosity for probes in cytoplasm is the effective viscosity, for which the molecular crowding plays key roles. Moreover, the actively nonequilibrium fluctuations such as the random force and the intracellular flow also impact the intracellular diffusion. With the experiments of lamellipodial fragments (*please see the new Figure S22*), we can exclude the role of intracellular fluid flow. After that, we think that the decreased intracellular diffusion in the lamellipodium of slow cells may be resulted from the raised molecular crowding in the cells. In another word, the experienced viscosity for the QDs has increased.

9. In Fig. 4a, the QDs were imaged with 3D defocus. In this configuration, both the positive and negative defocus generate the same off-focused pattern, right? How was the 35-nm axial resolution measured?

RESPONSE: This is a good question. We have implemented 3D single-particle tracking using a two-focal imaging system. We obtain the lateral coordinates (x, y) from the focused plane. The axial coordinate (z) is obtained from the defocused plane, which is generated by a lens (f = 300 mm) that is inserted into the light path. During 3D SPT, we generally focus near the bottom of cells which is near the surface of the glass. Consequently, in the defocused plane, the diffraction ring of a fluorescent probe is visible. The relationship between the diffraction ring radius and axial position would be determined before each experiment. The maximum detection range in the z direction is up to 2 μm . Since the thickness of the keratocyte lamellipodium is less than 1 μm (typically 200 nm for fast cells and 500 nm for slow cells), the relationship between the ring radius and the axial coordinates is reliable and the change of the diffraction ring radius is monotonic in our experiments.

For the axial resolution, it is determined by calculating the axial mean square displacement (MSD) of immobilized QDs on glasses, $MSD = 2\delta^2$, where δ represents the localization accuracy. Moreover, we previously compared the calibration relationships between particles in fixed cells and those on glass, and found they are similar. It indicates that the different refractive indexes between the cell and water have no obvious influence on the measurements of axial movements in our experiments (Jiang et al., *iScience*, 25, 104210 (2022)).

To make this clear to the readers, we have added more information in the Methods section for 3D single-particle tracking on page 24:

The maximum detection range in the z direction is up to 2 μm , which covers the thickness of the keratocyte lamellipodium that is less than 1 μm . Moreover, the different refractive indexes between the cell and water have been proven to have no obvious influence on the detection of z displacement in our platform⁵⁵. The axial resolution was approximately 35 nm, as was determined by calculating the axial MSD of immobilized QDs on glasses, with similar method as that in 2D.

10. Besides the change on the lamellipodium side, is there any change on the cell nuclei size and height in Fig. 4?

RESPONSE: We thank the review for the question. We have measured the cell nuclei size and height by 3D confocal microscopy, and found that there is no obvious difference between the fast- and slow-moving cells. The results are shown in the new Figure S9.

11. P15 “Interestingly, however, when the cell approaches another cell, the intracellular diffusion inside the cell becomes left-right asymmetric (Fig. 5c, d). We then use 3D SPT to measure the lamellipodium thicknesses more directly and find that the thicknesses are 245 nm in the inside part and 318 nm in the outside part (Fig. 4f, g and Fig. S15).” The cells in Fig.5 are in different states from those in Fig. 4. I suggest the authors to further measure the lamellipodium thicknesses of the cells in Fig 5a and Fig. 5c.

RESPONSE: This a nice suggestion. We have measured the lamellipodium thicknesses on both sides, of migrating cells in persistently moving and in turning, as in Figure 5a and 5c. The results are now shown in the revised Figure 4g, that the thicknesses are 245 ± 80 nm in the inside part and 318 ± 97 nm in the outside part for the turning cells, while they are 273 ± 91 nm in the inside part and 256 ± 66 nm in the outside part for the persistently migrating cells. It indicates that the left-right symmetry of the lamellipodium thickness has been broken in turning cells, with the outside part of lamellipodium being thicker.

Accordingly, we have revised the manuscript on page 15:

We then use 3D SPT to measure the lamellipodium thicknesses more directly and find that the thicknesses are 245 ± 80 nm in the inside part and 318 ± 97 nm in the outside part of the turning cells, while they are similarly in both sides (273 ± 91 and 256 ± 66 nm) for persistently migrating cells (Fig. 4f, g).

Figure 4. Measurements of lamellipodium thickness and a proposed mechanism for cell migration. **a–c**, A schematic of the 3D SPT setup. The axial position of QD is determined by its diffraction ring radius from the off-focused channel (**a**). A thick lamellipodium has a larger fluctuation of diffraction ring radius of diffusing QDs (**b**, **c**). **d**, Distributions of diffraction ring radius of QDs in fast and slow cells. **e**, The measured lamellipodium thicknesses of fast ($n=15$) and slow ($n=17$) cells. **f**, Distributions of diffraction ring radius of QDs in the inside and outside of lamellipodium of a turning cell. **g**, The measured lamellipodium thicknesses at the inside and outside parts of persistent ($n = 10$) and turning ($n = 10$) cells. **h**, **i**, Typical AFM images of fast (**h**) and slow (**i**) moving cells. The inset images are the corresponding CMFDA images of the cells. The black boxes on the inset images indicate the region measured by AFM experiments. **j**, Statistics of height in the front and the rear parts of lamellipodium (fast, $n = 5$; slow, $n = 8$). Inset: Height profile along the dashed line from the lamellipodium edges to the inside of the lamellipodium (along the arrow direction). $**P < 0.01$; $***P < 0.001$; NS, not significant. The boxes represent SD percentiles, where the whiskers represent data extremes and triangles represent the averages. **k**, Proposed mechanism for dynamic switching of a keratocyte cell between the fast and the slow migration modes. The cell in each mode is presented in both top and side views with the actin filaments inside the lamellipodium in orange. Black dots and lines with arrows represent the QDs and their trajectories.

12. It is recommended that both the data and the code are uploaded to a data deposit for open science.

RESPONSE: Thanks. We have uploaded the data in the revision, and shared the code on github (<https://github.com/chaojiang1901/3D-particle-tracking-MATLAB>).

RESPONSE to Reviewer #2:

This manuscript of Jiang and colleagues aims to address an interesting and important biological question on the role of intracellular diffusion in the control of cytoskeletal dynamics and cell migration. The authors provide experimental data demonstrating that spontaneous changes in the intracellular diffusion correlate with the density of actin filaments in lamellipodia, and hence, the speed of keratocyte migration. By tracking quantum dots taken up by the cell, the authors mapped out the diffusion speed across the cell and found that the diffusion of the quantum dots is faster in lamellipodia and slower in the cell body. By using an elegant combination of imaging techniques and pharmacological manipulations, the authors demonstrated that the suppression of intracellular diffusion results in thicker and less protrusive lamellipodia, while fast migrating cells exhibit high diffusion rate. Finally, the authors modulated the local diffusion rate by applying PEG and showed that local decrease of intracellular diffusion rate causes cell turning. Although this idea that intracellular diffusion regulates cell migration is extremely interesting to the field of cell biology and the data presented in the manuscript are novel, the experimental evidences supporting the main statements of the manuscript are often limited and inconclusive. My specific concerns and comments are as follows.

RESPONSE: We thank the reviewer for the support on our manuscript, and for pointing out the remaining issue.

1. Because several important conclusions were made based on the quantum dot tracking assay, it is essential to show that the quantum dots are indeed diffusing freely in the cytoplasm, but not trapped in the vesicles or transported as cargos.

RESPONSE: We thank the reviewer for the excellent suggestion. Indeed, it is very important to assure that quantum dots are freely diffusing in the cytoplasm and not trapped in the vesicles. We have several evidences to demonstrate the quantum dots are in the cytoplasm. First, if the QDs are trapped in vesicles, the intensity of the spot could be much higher as several QDs may exist in a vesicle. We find the intensities for the QDs in cells and for the individual QDs on glass are the same, suggesting the intracellular QDs are single particles (*please see the added Figure S2a, b*). Second, by co-culture of QDs with the cells for 2 hours, the QDs are endocytosed and trapped in the vesicles. In this case, we find that all the QDs are localized in the cell body, and they tend to form bright puncta with confined motion, which is obviously different from the diffusing QDs (*please see the added Figure S2c, d*). Third, if the fluorescent probes are trapped in the vesicles, the diffusion dynamics would be dependent on the vesicles, but not correlated to the probe size. However, we introduce the fluorescent dextrans with different molecular weights, and measure their diffusion coefficients. It is shown that the 70-kD dextrans diffuse much faster than the 500-kD ones, indicating a positive correlation between the probe size and their diffusion rate. And their diffusion rates are also different from that of QDs (*please see Figure S8*). These results indicate that the introduced probes by using the osmotic lysis of pinocytic vesicles are in the cytoplasm but not in the vesicles. Fourth, our measured diffusion rate for QDs is

similar with that in the previous study in which the QDs are introduced into the cytosol of keratocytes via the electroporation (Keren et al., Nature cell biology, 11, 1219 (2009)).

Moreover, to exclude the potential few aggregations of QDs in cytoplasm, we use an intensity filter to remove the spots whose intensities are beyond 2 times that of the mean intensity of all the QDs.

To be more clear to the readers, we have provided more experiment results shown in the new Figure S2, and added more words in the manuscript on page 5:

These QDs are proven to be located in the cytosol but not trapped in the vesicles (Fig. S2).

Fig. S2. QDs are dispersed in the cytosol but not trapped in vesicles. a, Fluorescent images of the loaded QDs in the cytosol via the osmotic lysis of pinocytic vesicles, as well as the QDs locating on the cover glass. The cell boundary is marked by the yellow line. **b,** QDs in the cytosol have the same intensities as those individually immobilized on glasses. **c,** Fluorescent images of endocytic QDs in cells. **d,** The diffusion coefficient of QDs in the cytosol is three times larger than the endocytic QDs trapped in vesicles. The boxes represent SD percentiles.

2. According to the authors, the data presented in Fig. S3 show that the intracellular actin flow does not have a detectable effect on the rate of quantum dot diffusion. Considering similar size of the quantum dots and pores in lamellipodia actin network, it is hard for me to believe that the quantum dots are not impacted by the flow. This concern is further supported by data presented in Fig. 2 of the manuscript, which show a significant decrease in the intracellular diffusion rate upon actin depolymerization with LatA. Comprehensive analysis of quantum dot diffusion under the experimental conditions that suppress the actin flow (simultaneous inhibition of actin polymerization and myosin contractility and/or in cells fixed with PFA) would strengthen the manuscript.

RESPONSE: We thank for the insightful question and the helpful suggestion. We have shown that the intracellular flow has no detectable influence on our measurements of QD diffusion (*please see Figure S4*). As mentioned by the reviewer, considering the similar size of the quantum dots and pores in lamellipodia actin network, the diffusion quantum dots would be affected by the flow. Here, we fully agree with the reviewer that our previous statement is not accurate and clear enough, since the influence of intracellular flow on intracellular diffusion is always existing. Our explanation for this discrepancy is that the intracellular flow is much slower than the motion of diffusing QDs, and the flow is not within the scope of our measurements. It is reported before that the actin flow of keratocytes is about 0.2 $\mu\text{m/s}$ (Wilson et al., Nature, 465, 373 (2010)), which is closed to the speed of cell migration. And the intracellular fluid flow is about 0.1 $\mu\text{m/s}$ (which is 40% of cell speed, reported by Julie lab's work, e.g. Keren et al., Nature cell biology, 11, 1219 (2009)). In our measurements, the diffusion rates of QDs are determined by linear fitting of the three points of MSD, the third point of which corresponds to a lag time of 120 ms. In such short time, the displacement of the actin flow and fluid flow are estimated to be 20 nm and 10 nm, respectively, which are smaller than our detection accuracy for single-particle tracking (~ 30 nm). Therefore, we cannot detect the effect of intracellular flow on the QD diffusion, which is also consistent with the previous study (Keren et al., Nature cell biology, 11, 1219 (2009)). With the same reason, the measured diffusion rates of QDs remain the same from the cell frame to the lab frame, as was shown in the Figure S3.

Another possibility is that the cytoarchitectural changes in the actin meshwork could hinder the intracellular diffusion by increasing the trapping or nonspecific binding of the probes. However, these above two hypotheses are ruled out, by evidences from the commonly reduced diffusion coefficients for other fluorescent probes with smaller sizes or different biochemical properties (Fig. 2h and Fig. S8).

Furthermore, we follow the reviewer's advice and performed new control experiments to suppress the actin flow with three kinds of drugs: latrunculin A (LatA) for perturbing the actin polymerization, blebbistatin (Bleb) for inhibit myosin II activity, and jasplakinolide (Jasp) for stabilizing the actin filaments. Previous study has shown that the combination of Bleb and Jasp can immobilize the actin network, which is not achieved by either drug alone. We find that, when treated by the combination of 50 μM Bleb and 1 μM Jasp, or 50 μM Bleb and 10 nM LatA, the cell migrations are slowed, as expected (*please see the added Figure S13*). Moreover, we observe the reduced intracellular diffusion of QDs, as well as the QD-excluded region in the diffusion map at the rear part of lamellipodium, suggesting the cells have entered the slow mode.

Together, since the speed of intracellular fluid flow is slower than the lower limit of our detection, the flow had no detectable influence on the measured intracellular diffusion rates. And our results suggest that by suppressing the actin flow or perturbing the actin polymerization, the cell migrations would be slowed and switch to the slow mode. But the reduced intracellular diffusion cannot be attributed to the reduced actin flow or the increased trapping effect of the actin meshwork in the slow-mode cells. It is more likely to be the

consequence of the 3D lamellipodium reorganization during the switch of migration modes, in which the molecule crowding is enhanced.

We have revised the manuscript accordingly.

On page 5: Moreover, we find that the speed of intracellular flow is slower than the lower limit of our detection, thus has no detectable influence on the measured intracellular diffusion rates (Fig. S4)²².

On page 9: Moreover, previous study has shown that the actin flow can be effectively suppressed by the simultaneous inhibition of action polymerization and myosin contractility³². We find that the treatments with Bleb and LatA, or with Bleb and the actin stabilization drug jasplakinolide (Jasp) show similar results as those for slow cells (Fig. S13).

Fig. S13. Similarly reduced intracellular diffusion rates and the diffusion maps with a QD-excluded region are both observed in cells after simultaneous inhibition of actin polymerization and myosin contractility. **a**, Average diffusion rates and migration speeds for cells with fast speed ($n = 6$), and for cells treated with 10 nM LatA and 50 μM Bleb ($n = 6$), or with 1 μM Jasp and 50 μM Bleb ($n = 9$). **b**, **c**, Diffusion maps of QDs show the QD-excluded region at the rear lamellipodia of the cells treated with Lat and Bleb (**b**), or with Jasp and Bleb (**c**). Data are shown as mean \pm SD.

3. The data presented in Fig 2 demonstrate that thicker regions of keratocytes (cell body and swollen lamellipodia) are characterized by the reduced intracellular diffusion rate. Can such difference in the diffusion be attributed to the movement of quantum dots in Z-direction?

RESPONSE: We appreciate the insightful question raised by the reviewer. We have characterized the 3D diffusion of QDs as shown in Figure S20. Although the axial diffusion has been enhanced in the thicker lamellipodium of slow cells, it is still significantly slower than lateral diffusion. It is the lateral diffusion that dominates the diffusion in the lamellipodium. By fitting the MSD curves in the axial direction, the diffusion coefficients in the Z-direction for fast cells and slow cells are determined to be 0.064 and 0.18 $\mu\text{m}^2/\text{s}$, respectively. Furthermore, as depicted in Fig. 2e, the 2D diffusion in the lateral (xy) directions are 1.53 $\mu\text{m}^2/\text{s}$ in fast cells and 0.75 $\mu\text{m}^2/\text{s}$ in slow cells. Therefore, the significant difference in the diffusion rates between fast and slow cells cannot be attributed to the movement of QDs in the Z-direction. Moreover, a positive

correlation between the cell migration speed and 3D intracellular diffusion still exists (Figure S21c), similar as the positive correlation from 2D measurements shown in Figure 11.

To be clear, we have added more descriptions about the diffusion in z-directions, in the revised manuscript on page 14:

Moreover, with the 3D SPT, the positive correlation between the cell migration speed and 3D intracellular diffusion rate still exists (Fig. S20), excluding the impact of axial diffusion on our results.

4. In Fig 3 & 5, the authors attempted to use PEG to alter intracellular diffusion by applying a solution with high osmotic pressure. Other than the change in the diffusion maps, are there any other accompanying changes would support the effect of PEG on the cytoplasm crowdedness and membrane tension? For example, in Fig. 5h, I would expect to see some quantifiable local deformation on side of the cell where PEG was applied, but the cell shape seemed to be unchanged. Additional experimental data are also required to rule out the effect of PEG on cell migration through the membrane tension.

RESPONSE: This is a good suggestion. As suggested, we have provided more experimental results for the cell migration regulated by extracellular osmotic pressures. First, in the revised Figure 5, after enhancing the contrast of original cell images, we are able to observe a noticeable shrink of the leading edge on the side of the cell where PEG is applied. To quantify the deformation of the cell shape, the cells are divided into the inside and outside parts along the central line parallel to the cell migrating direction (shown in the revised Fig. 5k, inset). Then we calculate the area ratio between the inside and outside areas for the cell at three time points (before, during, and after turning). It is shown that, the area ratio decreases when a cell approaches the PEG region and make a turn, and then the ratio goes back to 1 when the cell moves away from the PEG region. It suggests that PEG treatment locally shrinks the cell edges. Second, in the added Figure S9, we have measured the cell volumes after PEG treatment. It is seen that cell volume indeed reduces in the hypertonic solution, indicating that the cytoplasm crowdedness is increased. Third, in revised Figure S18, we further analyze the cell morphology when the cell moving mode is tuned by the change of extracellular osmotic pressures. It is shown that the cell area decreases in the hypertonic environment by adding PEG, and starts to increase in the hypotonic environment by adding water.

From above results and after careful consideration, we realize that the influence of PEG on intracellular crowding and membrane tension should be both considered in our study, since the membrane tension changes are coupled with cell volume changes during the osmotic adaptation (Roffay et al., PNAS, 118, e2103228118 (2021)). In our experiments using PEG, we have observed both the decreased cell volume and the deformed cell boundaries, which suggests a decrease in the membrane tension. Moreover, the membrane tension is associated with the cytoskeleton forces (Lieber et al., Current Biology, 23, 1409 (2013)). Our experiments that disturb the actin polymerizations and actomyosin contractions would affect the membrane tension (Figure 2

and 3). In addition, our another experiment that the cells are slowed by mechanical loading at the rear of the cell would lead to an increased membrane tension (Figure 3d and Figure S16). These results suggest that the membrane tension cannot be excluded in our experiments. However, its roles in the cell migration modes cannot be determined yet, without accurate measurements of the membrane tension by professional techniques such as the tether-pulling assay (Lieber et al., *Current Biology*, 23, 1409 (2013)). As the membrane tension is gradually recognized as an important mechanical regulator of cell migration, we think that the role of membrane tension in the switch of migration modes is an interesting and open question in our future investigations.

In this manuscript, we mainly focus on the association of 3D lamellipodium and intracellular dynamics during cells switching their migration modes. In the revised manuscript, we have added more discussions on the membrane tension and other possible mechanisms in the Discussion section on page 17 and 18:

The deformation of 3D lamellipodium is consistent with the reorganization of actin networks in slow cells, in which a denser actin network in front lamellipodia was shown by STED imaging (Fig. 2d) and by the electron microscopy recently¹⁸. It is reasonable to think that the actin network plays a role in the regulation of lamellipodium shape here^{3, 8, 32}. Moreover, within the front lamellipodium of slow cells, the increase of actin density together with the compartmentalization and concentration of large molecules, will increase the degree of molecular crowding locally and hence reduce the intracellular diffusion rates^{31, 40, 41}, considering that the cell volume remains similar (Fig. S9). To understand the coordination of 3D lamellipodium structure and intracellular diffusion with the cell migration behaviours, we propose a mechanism that involves the front-localized actin polymerization and increased molecular crowding in the lamellipodium.

The intracellular fluid flow is also an important factor in cell migration^{22, 35}. To test this, we study the migration of lamellipodial fragments, in which the intracellular fluid flow is changed due to the absence of myosin contractions at the cell rear³⁴. We find that a treatment of Bleb has no obvious effect on fragment speed and properties, which is consistent previous results⁶⁰. However, when treated by LatA, the fragments are slowed down, and more importantly, both lamellipodium deformation with swollen-up front and thinned-down rear, and the compartmentalized macromolecule diffusion are observed (Fig. S22), similarly as in the case of keratocyte cells in the slow mode. This experiment has eliminated the roles of intracellular fluid flow in the intracellular diffusion and in the regulation of cell migration modes here. Moreover, we note that the mechanism for switching of cell migration modes cannot be limited to the front-localized actin polymerization, but the possible role of membrane tension should also be considered^{61, 62}, as it associates with the cell volume, cytoskeleton forces and mechanical loading in our experiments. Besides, other biophysical mechanisms such as the mismatch between actin polymerization and treadmilling^{3, 63}, possible roles of molecular players aside from the actin^{16, 32, 64}, or hydrodynamic forces inside the cells^{23, 65}, may also be involved in the lamellipodium structure and intracellular diffusion during the migration switch. This remains to be resolved in the future.

We also revised the manuscript on page 15: Interestingly, when one side of the lamellipodium of the cell is entering the hypertonic region, the cell makes a turn towards the other side (Fig. 5j), with a local deformation of the lamellipodium where PEG is applied (Fig. 5k). ...

Figure 5. Turning cells show left-right asymmetries in both intracellular diffusion and lamellipodium structure. **a–d**, A cell (denoted as Cell A) migrates upwards and then make a turn to the right (**c**). The intracellular diffusion map in cell A changes from a normally left-right symmetric pattern (**b**) to an asymmetric one (**d**). Scale bar, 10 μm . **e**, Comparison of MSD curves of diffusing QDs in the inside and outside halves of lamellipodium in persistently migrating cells ($n = 5$) and in turning cells ($n = 9$). **f**, Comparison of the QD diffusion rates. Error bars indicate the SEM. *** $P < 0.001$; NS, not significant. **g**, Relative intensity of CMFDA in the lamellipodia of persistently migrating ($n = 12$) and turning cells ($n = 11$), along the cell outline from outside to inside, as indicated by the white line in the CMFDA image of a turning cell (insert). The solid lines and the shaded regions represent mean \pm SE. **h**, A cell changes its direction in response to the left-right asymmetric osmotic pressure. A glass microneedle is placed near the cell and flow 1% PEG to increase the local osmotic pressure. **i**, The hypertonic region is indicated by the cell-impermeant Fluo-4 pentapotassium salt that is mixed with PEG. **j**, Locations and orientations of the cell at different times. **k**, The area ratio of inside and outside parts of lamellipodium is plot as a function of time index (before (1), during (2) and after (3) turning) for three cells.

Minor suggestions

1. Most diffusion maps were created using data from a single cell. Since fish keratocytes have a relatively stereotypical cell shape, it makes sense to align maps from several similarly shaped and sized cells, and then compute the average of all the maps. This would help readers understand the phenotype at the population level.

RESPONSE: This is a nice suggestion. To produce the aligned diffusion maps, we collect thousands of trajectories in the cell frame of reference, from 5 cells with similar shape and size. And then we compute the averaged diffusion map from all the trajectories. It is shown that the averaged diffusion map of fast cells occupies the whole space in the lamellipodium, while the averaged diffusion map of slow cells still has a gap area in the rear lamellipodium and the overall diffusion rates are reduced. These results are consistent with single-cell diffusion maps in Figure 2. This new result as shown below is given in the added Fig. S6.

Accordingly, we have revised the manuscript on page 6:

In fact, the excluded gap region at the rear lamellipodium is still clearly observed in an aligned diffusion map from different cells, demonstrating that the compartmentalization is general in slow cells (Fig. S6).

Fig. S6. Diffusion map of tracks aligned from different cells with similar sizes and shapes. The diffusion map of fast cells (left panel, 5 cells, 4231 trajectories) is continuous, whereas that of slow cells (right panel, 5 cells, 2561 trajectories) has decreased diffusion rates with a gap area near the cell body where QDs are excluded.

RESPONSE to Reviewer #3:

In this paper, Jiang and colleagues study how fish keratocytes regulate their migration speed and direction by switching in a reversible manner their lamellipodium structure and associated intracellular diffusion. The authors identify that keratocytes can spontaneously switch from a fast to a slower mode of migration. The latter mode is associated with deformation of the lamellipodium from a flat configuration to a swollen front and thinned rear as evidenced by fluorescence microscopy, AFM and 3D single nanoparticle tracking. This migration switch is also correlated with a reduced intracellular diffusion and a compartmentalized macromolecular distribution within the lamellipodium as shown using single quantum dots tracking and actin distribution. Furthermore, using treatment with pharmacological drugs affecting cellular speed, or micromanipulation using microneedles and cellular situations restraining cell speed, the authors demonstrated this newly identified migration speed switch and correlated changes in lamellipodium structure and intracellular diffusion to be reversible. The authors also demonstrated that the migration switch is also involved during cell turning when it occurs only on one-half of the cell supporting a left-right symmetry breaking. Finally using solution to control locally the osmotic pressure, the authors propose that a molecular-crowding mechanism could be controlling the migration switch.

Overall, my opinion about the findings described in this manuscript is extremely positive. The novelty of this manuscript resides in identifying a new migration mode associated with changes in 3D lamellipodium structure and intracellular space organization. This is particularly striking as it was identified in fish keratocytes, one of the most studied cell model for cell migration. A noteworthy result is the starting point finding of this study, describing a positive correlation between cell migration speed and the intracellular diffusion probed by single nanoparticle tracking within the lamellipodium. This finding drove the authors to determine what changes in the organization of lamellipodium 3D architecture and intracellular space could be the basis of such correlation. In comparison to previous studies from Theriot's lab linking keratocytes speed and 2D shape, the present study introduced the variation of lamellipodium thickness as a crucial parameter controlling cellular speed. The introduction of 3D lamellipodium architecture, parameter that has been overlooked in the case of very flat keratocytes provides to this study, novelty and significance to the field of cell migration.

RESPONSE: We thank the reviewer for the positive and encouraging comments on our work.

Nevertheless, a regret about this manuscript is that molecular mechanism proposed to control the migration switch in this study is not yet convincing. Most evidences about a molecular-crowding mechanism rely on osmotic pressure manipulation known not only to affect molecular density within the lamellipodium, but also membrane tension, an important parameter also described as controlling speed and morphology of keratocytes (Keren lab and Theriot lab). Furthermore the authors do not provide alternative biophysical mechanisms

(mismatch between actin polymerization and threadmilling) or molecular players aside from actin that could regulate the control of lamellipodium thickness and macromolecular compartmentalization. In that sense, I feel that it would be informative to determine whether the migration speed switch or variations of lamellipodium thickness and/or intracellular diffusion are also observable for cell fragments that lack their cell body. At least, this would definitively exclude the described role of intracellular fluid flow in Keren et al, Nature Cell Biology 2009.

RESPONSE: Thanks for the very meaningful comments. After we thoroughly consider the experiments on osmotic pressure, we fully agree with the reviewer that the influence of PEG on intracellular crowding and membrane tension should be both considered. The membrane tension is known to be coupled with the cell volume during osmotic adaptations (Roffay et al., PNAS, 118, e2103228118 (2021)), and also the membrane tension has been proven to act as an important parameter in controlling the speed and morphology of keratocytes (Lieber et al., Current Biology, 23, 1409 (2013)). In the new experiments using PEG, we have observed both the decreased cell volume and the deformed cell boundaries (Figure S9 and Figure 5), which suggest a decrease in the membrane tension. Moreover, as the membrane tension is associated with the cytoskeleton forces (Lieber et al., Current Biology, 23, 1409 (2013)), our experiments that disturb the actin polymerizations and actomyosin contractions would affect the membrane tension (Figure 2). In addition, our another experiment that the cells are slowed by mechanical loading at the rear of the cell would suggest an increased membrane tension (Figure S16). These results suggest that the membrane tension may be involved in controlling the cell migration speed in our experiments. However, we think that the role of membrane tension in the cell migration modes cannot be currently determined yet, without accurate measurements of the membrane tension by professional techniques such as the tether-pulling assay (Lieber et al., Current Biology, 23, 1409 (2013)). The role of membrane tension in the switch of migration modes is an interesting and open question in our future investigations.

We also thank for the very nice suggestion on using cell fragments, in which the intracellular flow is changed due to the absence of myosin contractions at the cell rear. By generating the keratocyte fragments through treating cells with 100 nM staurosporine as was done before (Aroush, Dikla Raz-Ben, et al., Current Biology, 27, 2963 (2017)), we studied the lamellipodium thickness and intracellular diffusion of cell fragments. Consistent with the results for whole keratocyte cells, the lamellipodium deformation with swollen-up front and thinned-down rear, and the compartmentalized macromolecule diffusion are both found in the slow-migrating fragments. Therefore, this experiment could help us to exclude the role of intracellular flow in regulating cell migration modes. Please find the results shown in the new Figure S22.

Here, our work mainly focuses on the association of 3D lamellipodium structure and intracellular dynamics during cells switching their migration modes. To explain the phenomenon of decreased diffusion rates in slow cells, the molecular-crowding mechanism seems more reasonable, since we have excluded other possible factors that is associated with the intracellular diffusion, including the intracellular flow and the

nonspecific binding of probes. In the slow cells, the denser actin network in front lamellipodia has been observed by STED imaging in our experiments and by electron tomography in a previous study (Mueller et al., Cell, 171,188, (2017)). Moreover, within the front lamellipodium of slow cells, the increase of actin density together with the compartmentalization and concentration of large molecules, will increase the degree of molecular crowding locally and hence reduce the intracellular diffusion rates. Nevertheless, we recognize that there may be alternative biophysical mechanisms or molecular players aside from actin that could regulate the lamellipodium thickness and macromolecular compartmentalization, as was kindly noted in the comment.

To be more comprehensive, we have added corresponding discussions on the intracellular fluid flow, membrane tension and other alternative biophysical mechanisms in the switch of migration modes, in the Discussion section on page 18:

The intracellular fluid flow is also an important factor in cell migration^{22, 35}. To test this, we study the migration of lamellipodial fragments, in which the intracellular fluid flow is changed due to the absence of myosin contractions at the cell rear³⁴. We find that a treatment of Bleb has no obvious effect on fragment speed and properties, which is consistent previous results⁶⁰. However, when treated by LatA, the fragments are slowed down, and more importantly, both lamellipodium deformation with swollen-up front and thinned-down rear, and the compartmentalized macromolecule diffusion are observed (Fig. S22), similarly as in the case of keratocyte cells in the slow mode. This experiment has eliminated the roles of intracellular fluid flow in the intracellular diffusion and in the regulation of cell migration modes here. Moreover, we note that the mechanism for switching of cell migration modes cannot be limited to the front-localized actin polymerization, but the possible role of membrane tension should also be considered^{61, 62}, as it associates with the cell volume, cytoskeleton forces and mechanical loading in our experiments. Besides, other biophysical mechanisms such as the mismatch between actin polymerization and treadmilling^{3, 63}, possible roles of molecular players aside from the actin^{16, 32, 64}, or hydrodynamic forces inside the cells^{23, 65}, may also be involved in the lamellipodium structure and intracellular diffusion during the migration switch. This remains to be resolved in the future.

Fig. S22. Lamellipodial fragments show similar changes in both the lamellipodium thickness and intracellular diffusion when their migration speed reduces. a, CMFDA fluorescence images for migrating cells with fast speeds, and for cells treated with 50 μM blebbistatin (Bleb) or 10 nM latrunculin A (LatA). All the fragments are shown to move upwards. **b,** Normalized intensity profiles of CMFDA along a line across the lamellipodium, from the cell body to its leading edge in different conditions. **c,** Speeds of fragments in different conditions. The boxes represent SD percentiles, where the whiskers represent data extremes and triangles represent the averages. Since the loss of myosin contractions, the treatment of Bleb has no obvious effect on the fragment speed and the lamellipodium thickness. Ctrl, $n = 18$; Bleb, $n = 5$; LatA, $n = 5$. **d, e,** Intracellular diffusion maps for 70-kD dextran in fast (**d**) and slow (**e**) fragments.

A strength of this study relies on the multiplicity and soundness of experimental approaches used to define 3D lamellipodium architecture and intracellular diffusion properties. This said, I feel that a too strong emphasis is given on the sole intracellular diffusion properties. This certainly emerges from the fact it is an expertise from the authors' lab and it is not clear yet what is the molecular mechanism at play. But in my opinion, intracellular diffusion and compartmentalization within lamellipodium are a consequence of the 3D lamellipodium reorganization during migration switch.

RESPONSE: We agree with the reviewer's opinion. The change of intracellular diffusion should be the consequence of the 3D lamellipodium reorganization during the switch of migration modes. Accordingly, we have revised our manuscript to avoid the strong emphasis on the sole intracellular diffusion properties. We also rewritten the proposed mechanism involving the front-localized actin polymerization and increased molecular crowding in the lamellipodium, to explain how cells spatiotemporally coordinate the intracellular diffusion dynamics and the lamellipodium structure in regulating their migration behaviors. Please see the revised paragraph in the Discussion section on page 17:

The deformation of 3D lamellipodium is consistent with the reorganization of actin networks in slow cells, in which a denser actin network in front lamellipodia was shown by STED imaging (Fig. 2d) and by the electron microscopy recently¹⁸. It is reasonable to think that the actin network plays a role in the regulation of lamellipodium shape here^{3, 8, 32}. Moreover, within the front lamellipodium of slow cells, the increase of actin density together with the compartmentalization and concentration of large molecules, will increase the degree of molecular crowding locally and hence reduce the intracellular diffusion rates^{31, 40, 41}, considering that the cell volume remains similar (Fig. S9). To understand the coordination of 3D lamellipodium structure and intracellular diffusion with the cell migration behaviours, we propose a mechanism that involves the front-localized actin polymerization and increased molecular crowding in the lamellipodium.

In summary, the findings described in this manuscript deserve publication in Nature communications. Nevertheless in the current form, the presentation of the results in a sequential manner is not concise and produce overwhelming repetitions as for instance results obtained for intracellular diffusion with quantum dots nanoparticles or fluorescently-tagged macromolecules (dextrans) that are described in different parts of the manuscript. Strikingly many information about the nanoparticles are not presented up-front but through out as for example their size or their inert biochemical properties. Before being fully accepted for publication in Nature Communications, I recommend that an effort should be done to simplify the presentation of the results starting with the title that is highly descriptive and not conclusive. In this case, the emphasis should be given to the identification of a migration switch with the associated changes on 3D lamellipodium shape and intracellular diffusion properties.

RESPONSE: Thanks for these nice suggestions. Accordingly, we have reorganized the structure of our manuscript to be more straightforward, and put the similar experiments together. And we also simplify the presentation to be more concise. For example, in the revised manuscript, the examination of the influence of probe sizes and inert biochemical properties on the diffusion measurements are described immediately after the results of the QD diffusion. Please find the revised part on page 6:

Moreover, to eliminate the possible effect of biochemical property and size of the probes on the intracellular diffusion, we use 500- and 70-kD dextrans which are widely known as non-specific probes^{39, 45}, with their sizes comparable to and smaller than the QDs, respectively. They both show similar positive correlation between intracellular diffusion coefficient and cell speed, as well as the compartmentalized diffusion in the front lamellipodia of slow cells (Fig. S8).

In addition, as suggested, we have changed the title of the paper to be more conclusive. The new title is “Switch of cell migration modes orchestrated by changes of three-dimensional lamellipodium structure and intracellular diffusion”

In the following are only minor remarks, advices or questions:

Concerning the AFM measurements, why the authors did not perform lamellipodium thickness measurements in live keratocytes and used fixed samples with all the risks associated?

RESPONSE: Thanks for the good question. It is because the relative slow scan rate of the AFM and the fast migration speed of keratocytes. To get the AFM height profile, the scan of a defined area for a cell takes over 1 minute. Since the migrating of keratocyte is very fast, the keratocyte moves far away from its initial position during the scanning. Thus, it is hard to precisely measure the profile of lamellipodium thickness of live keratocytes. Despite that the fixed cell may be different from the live ones, the AFM results for the lamellipodium thickness are consistent with the results from 3D single-particle tracking of live cells.

Concerning G-actin labeling, I was wondering whether cell permeabilization could lead to loss of G-actin present in the cytosol and bias the ratio measurement between f-actin and g-actin?

RESPONSE: This is an excellent question. For G-actin labeling, we performed the experiment in fixed cells. Indeed, the cell permeabilization could lead to loss of G-actin. We have tested that after 24 hours of permeabilization, there would be no fluorescence signals of G-actin. Therefore, to prevent the loss of G-actin, we have imaged the cells immediately after the permeabilization and fluorescent labeling. Considering the loss rate of G-actin from cytosol is similar across the lamellipodium, we think the loss would not affect the ratio of actin monomers between the front part and rear part of lamellipodium (Figure S12b).

To be clear, we have added this point in the Methods section on page 25: **To avoid the loss of G-actin, the labelling and imaging of G-actin are performed immediately after the cell permeabilization.**

In the introduction section, the sentence “Given that the active fluid flow exists, it cannot fully account for an intracellular transport to match the cell speed” is not easy to understand and need to be revised.

RESPONSE: Thanks. We have revised this sentence to **“The active fluid flow with relatively slow speed cannot account for an intracellular transport that matches the cell speed”**.

On page 6, in the sentence “in a fast cell, the diffusion is essentially heterogeneous across its whole lamellipodium”, it is not clear what the authors mean by heterogenous. Do they mean continuous or uncompartimentalized?

RESPONSE: Thanks for pointing out this. Yes, we want to describe the continuity of diffusion in fast cells. We revised this sentence to **“in a fast cell, the diffusion of QDs is present across the whole lamellipodium”**.

In figure 2, images of CMFDA distribution in panel 2c appear to display a very low contrast with background signals from outside the cell to be as intense as in the lamellipodium. It would be worth matching the background signal outside the cell for all 4 images.

RESPONSE: This is a nice suggestion. We have set the background signal outside the cell to 0 for all 4 images in the revised Figure 2. It is very helpful to enhance the contrast.

On page 8, in the sentence “the presence of strikingly excluded gap regions in the lamellipodia of slow cells is resulted from a physical barrier”, “is resulted” should be replaced by “is resulting”.

RESPONSE: Thanks. We have revised it.

On page 11, in the sentence “After 195 s when the cell is isolated from the population”, “isolated” should be replaced by “separated”.

RESPONSE: Thanks. We have revised it.

In the discussion (page 17), “reversible” should be replaced by “reversible”

RESPONSE: Thanks. We have revised it.

REVIEWERS' COMMENTS

Reviewer #1 (Remarks to the Author):

Overall, the authors have addressed my previous comments satisfactorily. For Q6, the authors tried to provide direct evidence, but due to technical limitations, microneedles could not follow the cell movement accurately and fast enough, which inevitably affected the cell movement. The authors provide an alternative solution: by observing that the dynamics of QDs distribution change rapidly when cells switch from fast mode to slow mode, the author then relates this process to physical squeezing rather than other biological causes, and the mechanism is further explained in the discussion section. With that, I think the manuscript can be accepted at its current form.

Reviewer #2 (Remarks to the Author):

The authors have made several key improvements to strengthen the manuscript. The revised manuscript includes convincing evidence that:

1. The quantum dots are freely diffusing inside the cell and not confined in vesicles;
2. Actin flow has limited effect on the diffusion of quantum dots;
3. The hypertonic solution induces a local change in cell shape.

I think at this point the authors have reasonably verified their conclusions and the current version of the manuscript is appropriate for publication in Nature Communications. I greatly appreciate the author's effort and for this interesting study.

Reviewer #3 (Remarks to the Author):

The authors have answered to all my comments positively. I now recommend their manuscript for publication in Nature Communications.

Nevertheless one point that I did not raise in my first review stroke me in this latter version. This point concerns the interpretation of the keratocyte turning process. Indeed the authors define the outside and inside points during turning and clearly describe that the outside point displays similar changes observed in the cells in the slow mode (increased thickness, decreased intracellular diffusion) while the inside point conserved the characteristics of cells in the fast motion. In that configuration, I think the authors do not explain clearly how these observations support the turning process. Indeed the speed imbalance between the outside point and inside point should lead to a turning in the opposite direction. Maybe my view is too simple, but I would advise the authors to discuss this mismatch and proposed a clearer explanation on how their observations could be interpreted to support the turning process geometry.

We thank the reviewer for the careful reviewing of our manuscript and the very helpful comments and suggestions. Accordingly, we have addressed all comments and questions point by point below. For convenience, the reviewers' comments are shown *in black*, our responses are shown *in blue*, and the corresponding changes in the manuscript are shown *in red* in the revised manuscripts.

RESPONSE to Reviewer #1:

Overall, the authors have addressed my previous comments satisfactorily. For Q6, the authors tried to provide direct evidence, but due to technical limitations, microneedles could not follow the cell movement accurately and fast enough, which inevitably affected the cell movement. The authors provide an alternative solution: by observing that the dynamics of QDs distribution change rapidly when cells switch from fast mode to slow mode, the author then relates this process to physical squeezing rather than other biological causes, and the mechanism is further explained in the discussion section. With that, I think the manuscript can be accepted at its current form.

RESPONSE: We thank the reviewer for the very careful reviewing and the positive comments.

RESPONSE to Reviewer #2:

The authors have made several key improvements to strengthen the manuscript. The revised manuscript includes convincing evidence that:

1. The quantum dots are freely diffusing inside the cell and not confined in vesicles;
2. Actin flow has limited effect on the diffusion of quantum dots;
3. The hypertonic solution induces a local change in cell shape.

I think at this point the authors have reasonably verified their conclusions and the current version of the manuscript is appropriate for publication in Nature Communications. I greatly appreciate the author's effort and for this interesting study.

RESPONSE: We thank the reviewer for the very careful reviewing and the support on our manuscript.

RESPONSE to Reviewer #3:

The authors have answered to all my comments positively. I now recommend their manuscript for publication in Nature Communications.

Nevertheless one point that I did not raise in my first review stroke me in this latter version. This point concerns the interpretation of the keratocyte turning process. Indeed the authors define the outside and inside points

during turning and clearly describe that the outside point displays similar changes observed in the cells in the slow mode (increased thickness, decreased intracellular diffusion) while the inside point conserved the characteristics of cells in the fast motion. In that configuration, I think the authors do not explain clearly how these observations support the turning process. Indeed the speed imbalance between the outside point and inside point should lead to a turning in the opposite direction. Maybe my view is too simple, but I would advise the authors to discuss this mismatch and proposed a clearer explanation on how their observations could be interpreted to support the turning process geometry.

RESPONSE: We thank the reviewer for the very careful reviewing, the positive comment, and the insightful question. Indeed, in turning cells, the outside of lamellipodium shows an increased thickness and decreased intracellular diffusion, while the inside part essentially remains unchanged. It could also be explained by the mechanism that we have proposed in the manuscript: the front-localized actin polymerization and increased molecular crowding in the lamellipodium. A cell cannot be regarded as an ideal rigid body, when the cell makes a turn, the outside of its lamellipodium undergoes shrinkage while the inside expands, as can be seen in Figure 5a,c and 5h-k. This means that the cell relies on an imbalance between its two sides for a biased motion, gradually turning towards the inside direction. Due to the decreased forward speed of the membrane on the outside, the actin network will become denser and the local molecular crowding become stronger, which would increase the thickness of the outside lamellipodium and decrease the local intracellular diffusion rate.

To make this clearer, we have improved the sentences in the discussion to explain the cell turning process.

“Similarly, in turning cells, the speeds between the two sides of lamellipodium are imbalanced that the outside of the lamellipodium undergoes shrinkage while the inside expands (Fig. 5a, c). Due to the decreased forward speed of the membrane on the outside, the actin network becomes denser and the local molecular crowding becomes stronger, which would increase the thickness of the outside of the lamellipodium and decrease the local intracellular diffusion rate.”